# Deep reinforcement learning can promote sustainable human behaviour in a common-pool resource problem

Raphael Koster [1,6] ✉, Miruna Pîslar[1,6] ✉, Andrea Tacchetti[1], Jan Balaguer[1], Leqi Liu[1,2], Romuald Elie[1], Oliver P. Hauser [3], Karl Tuyls[1], Matt Botvinick [1,4] & Christopher Summerfield [5]

A canonical social dilemma arises when resources are allocated to people, who can either reciprocate with interest or keep the proceeds. The right resource allocation mechanisms can encourage levels of reciprocation that sustain the commons. Here, in an iterated multiplayer trust game, we use deep reinforcement learning (RL) to design a social planner that promotes sustainable contributions from human participants. We first trained neural networks to behave like human players, creating a stimulated economy that allows us to study the dynamics of receipt and reciprocation. We use RL to train a mechanism to maximise aggregate return to players. The RL mechanism discovers a redistributive policy that leads to a large but also more equal surplus. The mechanism outperforms baseline mechanisms by conditioning its generosity on available resources and temporarily sanctioning defectors. Examining the RL policy allows us to develop a similar but explainable mechanism that is more popular among players.

A healthy economy is sustained by trust among economic actors[1–3]. For example, a buyer comes to trust that a supplier's goods are of expected quality[4,5]; an employer trusts that employees will provide adequate work[6,7]; and the state trusts its citizens to meet minimum standards of civic responsibility[8]. A canonical trust problem arises when resources are drawn down from a common pool and allocated to a group, who may then choose to replenish the pool with interest[9]. The search for mechanisms that encourage sustainable reciprocation in this class of common pool resource (CPR) problem has been a central concern in the social sciences[10,11]. Unfortunately, mechanism designers must overcome a classic social dilemma: when resources are offered, each recipient can choose to reciprocate (for the common good) or selfishly keep the proceeds without giving anything back (for individual benefit). In repeated settings, selfish recipients will maximize their own payoff in the short run, but it is in the collective long-term interest to ensure that obligations are met, in order to sustain future allocation[12,13]. For example, if company founders fail to repay a government start-up

loan, the government may be left with fewer resources to inject into the future economy; likewise, if employees shirk, a business may fail, leaving them unemployed. The same dynamic governs sustainable stewardship of shared resources, such as a financial endowment, harvestable stocks like forests or fisheries, or the global environment.

Since the pioneering work of Elinor Ostrom, solutions to this problem have emphasised the ways that people can self-organise to sustainably manage a shared resource[9]. In laboratory settings, these solutions have been studied by equipping players with auxiliary signals or actions that allow them to communicate, influence each other or self-organise, which can increase voluntary contributions towards a public good. For example, in multiplayer games, cheap talk[14,15] and onymity[16] sustain reciprocation, and participants will often opt-in to games with mechanisms that allow players to ostracise uncooperative group members at personal expense or that permit sanctioning schemes to punish free riders[17,18]. Cooperation tends to increase when players can vote for exclusion of free riders[19], observe who is

[1]Google DeepMind, London, UK. [2]Princeton University, Princeton, USA. [3]University of Exeter, Exeter, UK. [4]Yale Law School, Yale University, New Haven, USA. [5]University of Oxford, Oxford, UK. [6]These authors contributed equally: Raphael Koster, Miruna Pîslar. ✉e-mail: rkoster@google.com; mirunapislar@google.com

trustworthy[20], or enter into contracts that enforce minimum reciprocation levels[21,22]. Whilst insightful, this work leaves unaddressed the starker question of how resources can be allocated by a principal agent (or social planner) in ways that incentivise trust, when forms of institutional self-organisation that permit mutual sanctioning, voting, or contracting among recipients are unavailable. This is a potentially daunting problem, because the study of CPR games has typically shown that without such institutional coordination mechanisms, private contributions are unstable and prone to collapse. Here, we asked whether there exist top-down resource allocation mechanisms that can lead to a sustainable, inclusive economy, for example by encouraging recipients to reciprocate because they consider allocation to be beneficial, fair and transparent. This is a fundamental problem with widespread implications for the provisioning of public goods, and theories of optimal taxation, remuneration, and welfare.

The main innovation that we bring to tackle this problem is the use of new tools from AI research. Deep neural networks are powerful function approximators, allowing them to learn intricate policies that depend on a complex sequence of past events. Here, we asked whether a deep network could learn to dynamically allocate resources to human recipients in ways that encouraged them to sustain the common pool. A natural methodological toolkit for designing a resource allocation mechanism is deep reinforcement learning (RL), in which neural networks can be optimised to take actions that maximise a scalar quantity (an objective function or 'reward')[23]. Here, we define this objective as the funds which human players take home from the game, aggregated over rounds and individuals, so that games sustained for longer yield higher rewards. In other words, the RL agent's objective is to achieve high social welfare for the group across all rounds of the game.

In applying tools from AI research to study resource allocation, we are building upon earlier work, in which we used deep RL to maximise reported human preferences (votes) over a resource allocation policy in a linear public goods game[24,25]. This previous work made several simplifying assumptions that our new approach solves. First, previously the RL agent maximised votes, whereas our agent maximises actual long-term welfare of the group. Second, rather than use a pool of fixed size, now the pool size varies as assets are drawn down and replenished, and so the agent has to learn a policy that depends on the past history of contribution and the current level of resources (indeed, we shall see that the latter is a critical factor in the solution it discovers). To meet this challenge, we use a deep RL system that is equipped with a memory network allowing it to condition its policy on the history of the current game (rather than treating each round of exchange as if it were independent from every other).

To evaluate our RL model, we first chose a space of simple mechanisms, drawn from a continuum along which allocations depend to varying degrees on the level of reciprocation made by each player. These baseline mechanisms expose why the problem is theoretically interesting. At one extreme, the social planner could offer equal resources to recipients irrespective of their past reciprocation. However, this "equal" policy incentivises free riding, as self-interested recipients can rely on others to carry the burden of reciprocation without jeopardising their own relative future receipt[26,27]. Thus, some argue that unconditional welfare—or a universal basic income—from the state discourages citizens from seeking work[28]. At the other extreme, a social planner could offer recipients investments that are proportional to their past reciprocation, so that self-interested agents are encouraged to reciprocate in expectation of future receipt[29]. However, this "proportional" policy means that trustees receiving less will have reduced capacity to reciprocate, further diminishing their subsequent allocation – and thus locking them into a cycle of ever-diminishing resources. For example, cutting unemployment benefits may itself create circumstances unfavourable to reemployment[30], such as long-term ill-health[31], leading to workers being permanently excluded from the labour market.

In this work, our approach to identifying a sustainable allocation policy draws on ideas from game theory and cognitive science as well as AI research. We first collect data from a large group of humans playing a multiplayer trust game. Using machine learning techniques, we build an accurate simulation of human behaviour in the game, populated by neural networks that behave like human players. We then use RL to train an artificial agent to allocate resources to simulated people in a way that should maximise sustainable exchange. We then test the mechanism with real human participants. Surprisingly, the RL social planner identifies a resource allocation mechanism that successfully promotes sustainable behaviour among people, even without endogenous mechanisms that allows them to communicate or self-organise. We then investigate the properties of this mechanism, and build a simple, explainable heuristic that can recreate it, which we find to be equally successful at promoting sustainable behaviour.

## Results

We devised an infinitely repeated multiplayer trust game, based around the challenge of sustaining a common pool resource. The social planner is assumed to be an individual or institution that decides who gets what by allocating monetary resources to $p$ human recipients[32,33]. The allocation mechanism can either be designed by human hand or discovered by a reinforcement learning (RL) agent. On each round $t$ of the game, each recipient $i$ is allocated an endowment $e_{i,t}$ from a common pool with resources $R_t$ (so that $\sum_i e_{i,t} \leq R_t$) and freely chooses to make a reciprocation $0 \leq c_{i,t} \leq e_{i,t}$ back to the pool, with the remaining surplus $s_{i,t} = e_{i,t} - c_{i,t}$ retained for private consumption. The mechanism determines the level of resources that is allocated to each player, including no allocation. The pool is initialised to its maximum value $R_0$ and updated on each round so that $R_t = \min(R_0, R_{t-1} + \Delta R_t)$ with $\Delta R_t = -\sum_i e_{i,t} + (1+r)\sum_i c$ where $r$ is a growth factor. Imposing a maximum value on the pool reflects the assumption that in many settings (e.g. ecosystems, certain business models) resources cannot grow beyond a fixed carrying capacity. This abrupt nonlinearity can also reflect that individual business models, technologies or satiated markets can reach maximums of growth. The game continues for an unknown number of rounds (at which point any surplus funds in the pool are lost) or until the pool is fully depleted. The agent objective was to discover an allocation mechanism that maximises the aggregate surplus over rounds and players $\sum_i \sum_t s_{i,t}$. For all games described here, we used $p = 4$, $R_0 = 200$, and $r = 0.4$. We provide an illustration of the game in Fig. 1A, and an overall roadmap for our research project in Fig. 1B.

In Exp.1, we recruited an initial cohort of 640 participants (160 groups of 4 participants) who played the game online over 40 rounds (this number of rounds was a priori unknown to players; in our follow-up experiment below, we introduce a continuation probability to further reduce end-game effects). After the game, players received bonus payments proportional to their average surplus over a randomly and uniformly chosen number of rounds ($n \leq 40$; this rule, which was clearly explained to participants, discourages strategic responding based on when the episode will end). Of these groups, 120 played the game under pre-determined allocation mechanisms (hand-designed by the researcher). These were drawn from a space of baseline mechanisms that computed the allocation to player $i \neq j$ on round $t$ (for $t > 1$; allocation was always equal on the initial round $t_0$) as a weighted sum of a proportional and equal allocation policy, controlled by a mixing parameter $w$:

$$e_{i,t} = w \cdot \frac{R_t}{p} + (1-w) \cdot R_t \cdot \frac{c_{i,t-1}}{\sum_j c_{j,t-1}} \qquad (1)$$

In Exp. 1, 40 groups each played under baseline allocation mechanisms defined by $w = 0$ (*proportional*), $w = 0.5$ (*mixed*) and $w = 1$ (*equal*). Note that $w = 1$ in the equation above (i.e. all players receive an

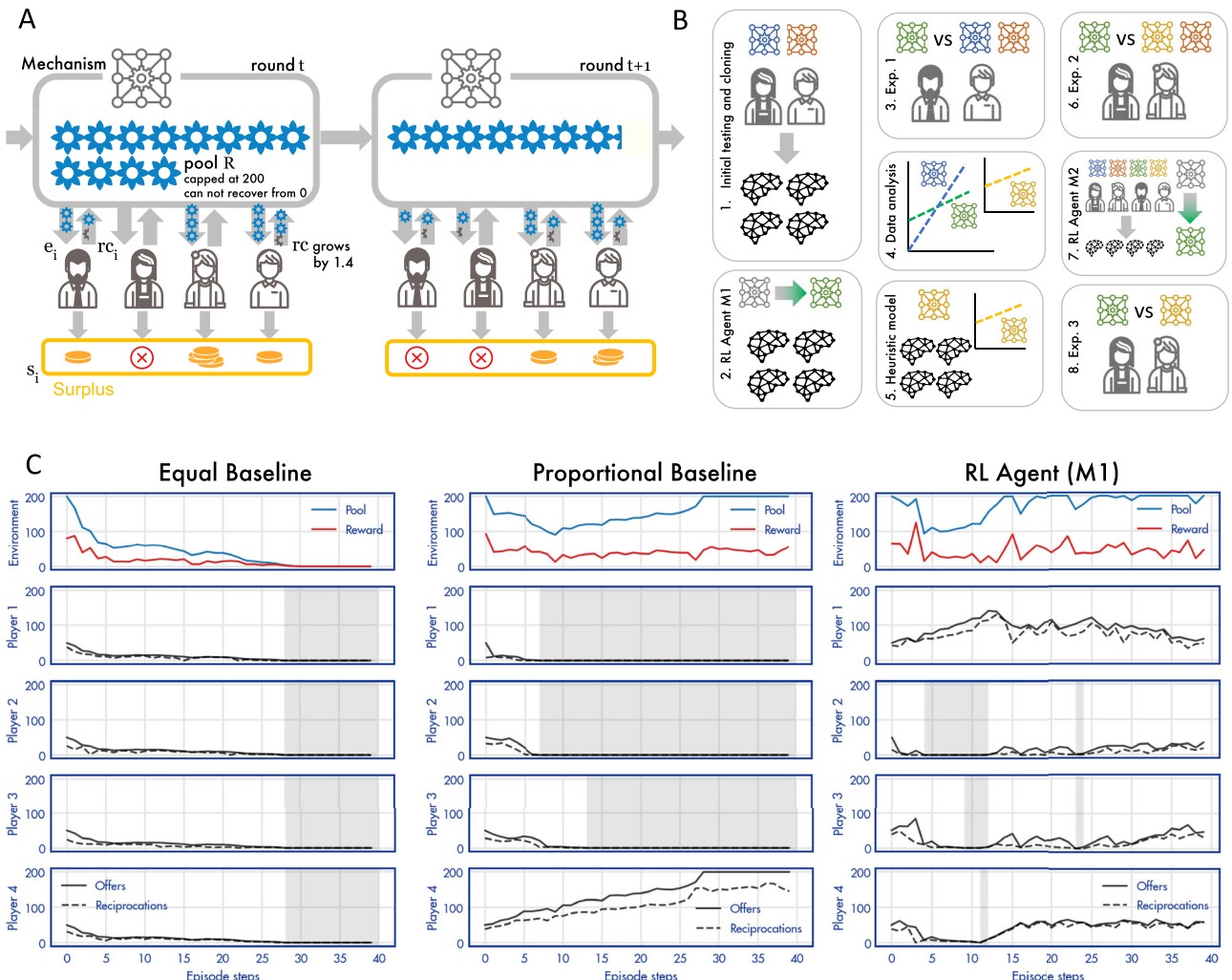

**Fig. 1 | Illustration of the game. A** Two rounds (denoted $t$) are illustrated (columns). In each, a mechanism allocates resources (blue flowers) from a pool of size $R$ to $p = 4$ players, who each choose a quantity to reciprocate, with any remainder going to surplus (gold coins). For example, in the schematic, in round $t$ the first player (left) receives 2 flowers, and reciprocates 1, generating a surplus of $r = 0.4$ (amount due to growth factor shown in grey). The pool size is depleted by the allocation and replenished by the reciprocations. Note that players who receive no resources cannot reciprocate (e.g. centre left player on round $t$) (**B**) Illustration of our approach. First, (1) we collected data from human participants under a range of mechanisms defined by different values of $w$, and used imitation learning to create clones that behaved like people. Then (2) we used these clones to train the RL agent, and (3) conducted Exp.1, in which we compared the RL agent to baselines. Next, (4) we analysed the RL agent policy, and constructed a heuristic approximation that was more explainable (the 'interpolation baseline'), which (5) we tested on behavioural clones, and (6) compared to the RL agent (and proportional mechanism) in Exp. 2. Finally, (7) we used all of the data so far to retrain a new version of the RL agent, and (8) compared it to the interpolation baseline in Exp. 3. **C** Example games (using behavioural clones). The game starts with $R_0 = 200$. Left: offers (full lines) and reciprocations (dashed lines) to four players (lower panels) over 40 trials (x-axis) in an example game with the equal baseline. The grey shaded area indicates where each player receives an offer of zero. The top panel shows the size of the pool (blue) and the total per-trial surplus (red). The middle and right panels show example games under the proportional baseline and the RL agent, respectively. Note that in the example proportional baseline game, three players fall into poverty traps, leaving a single player to contribute, and increasing inequality.

equal share from the social planner) creates the largest incentives for free riding. In Fig. 1C we show dynamics for the pool (top panels) and each player (lower panels) for an example game (with virtual players) in equal and proportional baseline conditions (left and middle panels). Equal allocation most often leads to a rapid collapse in reciprocation and thus in pool size, similar to that seen in linear public goods games (Fig. 1C, left), where the pool dwindles to zero and no further allocations can be made (so all players are "excluded"). Under proportional allocation (where any player that gives zero will receive no future allocations) we typically observe a pattern whereby several of the players are excluded early. For example, in the game shown in Fig. 1C (middle panel) three players drop out early in the game, leaving Player 4 to sustain the pool. Thus, as expected, proportional mechanisms create inequities when players fall into poverty traps, which leaves just

a single individual in the economy – removing the need for mutual trust, and highlighting the uncomfortable maxim that under such a scheme "the monopolist is the conservationist's best friend"[34].

Statistically, we observed that games played under the *equal* baseline led to lower surplus than other conditions (equal <mixed, $z = 4.58$, $p < 0.001$; equal <proportional, $z = 5.28$, $p < 0.001$; all two-tailed Wilcoxon rank sum tests at the group level unless otherwise specified), whereas games played under the proportional baseline led to higher Gini coefficient (the inequality of total player surplus by the end of the game) than other conditions (proportional > mixed, $z = 5.89$, $p < 0.001$; proportional > equal, $z = 4.76$, $p < 0.001$). In Fig. 2A (right panel) we show aggregate surplus and Gini coefficient for each mechanism in the games played with human participants in Exp.1. The equal (blue dots) and mixed (purple dots) conditions yield low surplus

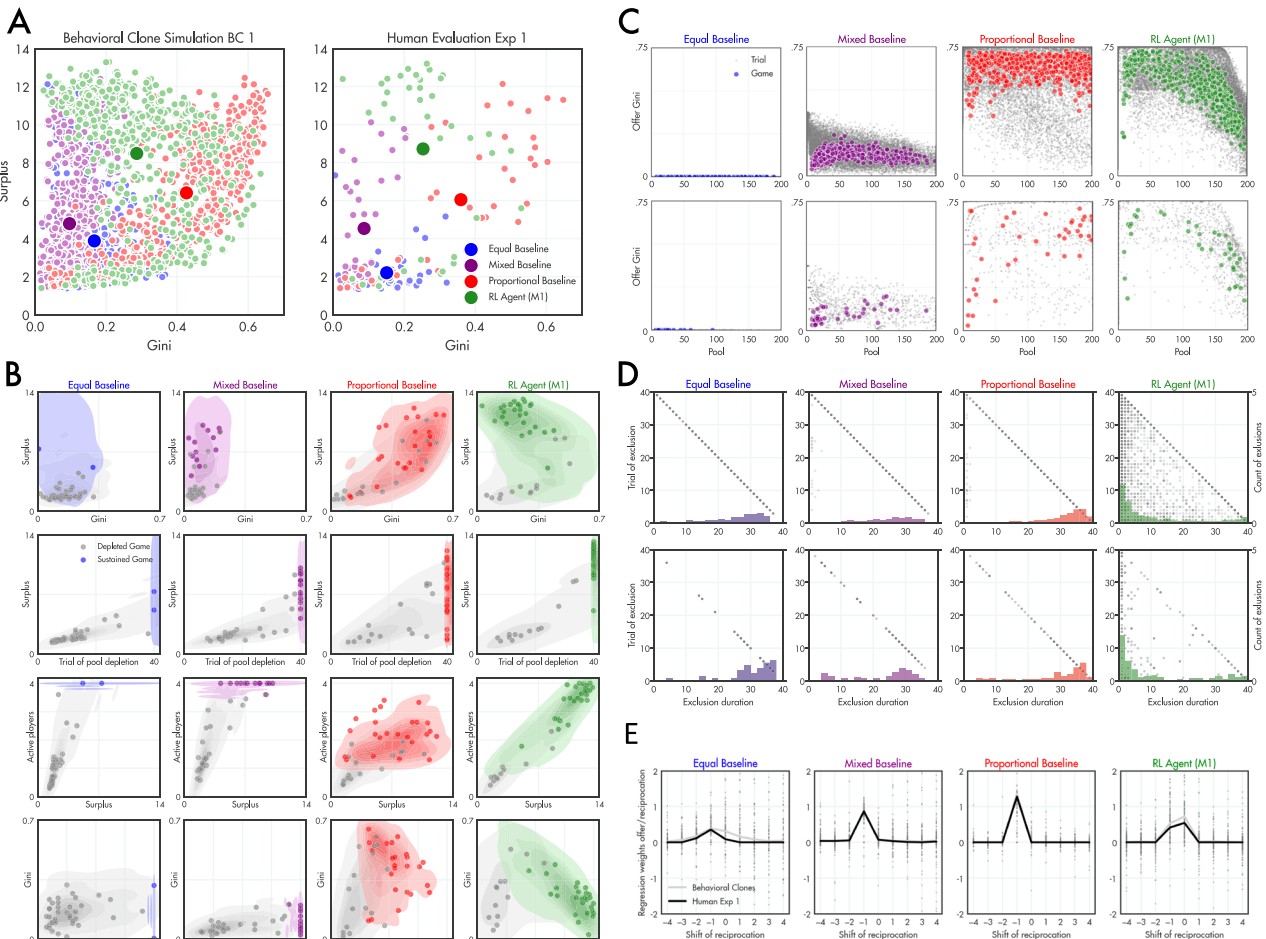

**Fig. 2 | Results of first trained mechanism against baselines. A** The surplus and Gini coefficient generated from games played under three baseline mechanisms (blue, purple and red) and the RL agent (green), for virtual players (left panel) and human participants (right panel) in Exp. 1. Each small dot is a game; the larger dot is the mean over games. **B** Correspondences between predicted outcomes (from virtual players, shading) and observed outcomes in Exp.1 (dots) for each baseline mechanism and the RL agent. Shown separately for games that were sustained to the end by at least one player (colours) and those where the pool was exhausted prematurely (grey). **C** The average Gini coefficient of the offer made to players as a function of the pool size, for individual trials (grey dots) and games (coloured dots),

both for behavioural clones (upper panels) and human data in Exp.1 (lower panels). **D** Exclusions occur when a player receives nothing for one or more consecutive rounds. Here, we plot the duration of exclusions against the trial on which they are initiated (dots). Points on the diagonal indicate that the player was never reincluded (exclusion lasts until trial 40). The superimposed coloured histograms show the count of exclusions for each duration bin (of width 2). Note that unlike baselines the RL agent excludes frequently, but for short durations. **E** The offer made by each mechanism to each player as a function of the lagged contribution of that player over adjacent trials. Dots are individual coefficients; black line is the median.

and low Gini (-0.1; because surplus is uniformly low), whereas proportional (red dots) has a higher surplus but incurs a much higher Gini of just under -0.4. When we computed the fraction of games that were sustained to the final (40th) round, we found that 60% of proportional games were sustained with at least one player, but none with all four players; by contrast, in mixed or equal conditions, where games were either sustained by everyone or not at all, 30% and 5% of games finished with all four players still active. Thus, our baseline mechanisms were not successful at encouraging sustained reciprocation from human players.

The final 40 groups of humans in Exp.1 played the game in the RL agent condition, where allocation decisions were made by an AI model that had been trained to maximise recipient surplus. To train the agent, we first collected several hundred games in which a different sample of humans played under a range of policies (baseline mechanisms with randomly sampled $w$). We then used imitation learning to create *virtual players*, which were recurrent neural networks, and whose behaviour was optimised to be as similar as possible to that exhibited by humans in the training cohort. We then combined deep policy gradient methods (25) with graph neural networks (26) to train an agent to take

on the role of social planner, optimising it to maximise the aggregate surplus over virtual players (see *Materials and Methods* for a full description of this pipeline, and Fig. 1C right panel for example games under the trained RL agent).

The mean surplus and Gini coefficients from simulated games (with virtual players) involving the RL agent are shown as the green dots in Fig. 2A (left panel), along with the corresponding empirical observations from real humans in Exp.1 (right panel). Strikingly, in Exp.1 the RL agent generated a surplus that was ~150% greater than the highest baseline (proportional) and did this under a much lower Gini of just over ~0.2 (Fig. 2A, green dot). Over games, the RL agent generated a higher surplus than the other conditions (*RL agent > proportional*, $z = 3.25$, $p = 0.001$; *RL agent > mixed*, $z = 4.38$, $p < 0.001$; *RL agent > equal*, $z = 6.31$, $p < 0.001$), and it also had a lower Gini coefficient than the proportional baseline (*RL agent < proportional*, $z = 2.78$, $p < 0.01$) but not than the equal or mixed conditions (Gini is lowest after a collapse in welfare, because nobody gets anything; *RL agent > mixed*, $z = 6.29$, $p < 0.001$; *RL agent > equal*, $z = 4.19$, $p < 0.001$). Overall, 65% of all games played under the RL agent were sustained to the end with at least one player, and 55% of these were with all four players. We

additionally generated two further indices of economic inclusivity: the average number of active players across the game (those receiving a nonzero offer) and the average trial on which the pool was depleted (or the maximum number of rounds [40], whichever was lower). The RL agent sustained the pool for longer than the equal ($z = 5.83$, $p < 0.001$) and mixed ($z = 2.37$, $p < 0.05$) but not proportional baseline; however it maintained more active players than both the equal ($z = 4.45$, $p < 0.001$) and proportional ($z = 3.16$, $p < 0.01$), but not mixed, baselines.

Our RL agent performed well with new participants because our simulated virtual players performed almost exactly like real human players of the game. This is most clearly visible in Fig. 2B where we plot, for Exp.1, the joint distribution of surplus, Gini coefficient, and the two inclusivity measures (active players and mean depletion trial). The dots in each plot show the data from Exp. 1 and the shaded area shows the distribution of data generated using behavioural clones: without exception, the overlap is striking (see also Fig. S1). This means that we have created a model of how people play the game that successfully generalises across different game variants, and could in theory be used to evaluate the likely success of any new mechanism.

The agent thus found a way to encourage humans to reciprocate collectively, producing a sustainable surplus without compromising equality. Of note, examining the relationship between surplus and Gini in Fig. 2B, we can see that for the proportional baseline, these are positively correlated both for games that are sustained to the end ($r = 0.69$, $p < 0.001$) and those that are not ($r = 0.89$, $p < 0.001$), meaning that surplus is always generated at the expense of equality; similar positive correlations were observed for mixed conditions when games ended prematurely ($r = 0.54$, $p < 0.01$). By contrast, under the RL agent mechanism, surplus is positively correlated with Gini for those games that end prematurely ($r = 0.81$, $p = 0.001$), but negatively correlated for those games that are sustained to trial 40 ($r = -0.5$, $p < 0.008$). Thus, under the mechanism discovered by the agent, games with higher surplus were more egalitarian, which is striking because the agent was not trained to maximise equality. In fact, simulations using virtual players predicted that games played under the agent mechanism would last an average of $271 \pm 251$ rounds, compared to $32 \pm 28$ and $105 \pm 102$ for the equal and proportional baselines respectively (Fig. S2).

What was the agent learning to do? Our baseline mechanisms do not condition allocations on the pool size, whereas our agent (which receives pool size as an input) could use this knowledge to flexibly scale its generosity to available resources. To test this, in Fig. 2C we plotted the Gini coefficient of the offer as a function of pool size for individual trials (grey dots) and the average over games (orange dots) for each of the mechanisms, both for virtual players (top panels) and human participants in Exp. 1 (bottom panels). As can be seen, whereas proportional offers were typically highly unequal, the RL agent tended to distribute more equally when resources were more abundant. A more direct test of the agent's policy is provided in a controlled experiment using virtual players, where we measure how its behaviour changed in response to spurious information about the level of resources in the pool. As implied by Fig. 2C, this analysis revealed that the agent was more punitive when the pool was low, but made more generous offers and distributed resources more equally as the common pool grew (Fig. S3). This behaviour is reminiscent of Kuznets theory, which proposes a curve describing how nations become more egalitarian as their economy develops[35].

One salient aspect of the proportional baseline is that a reciprocation of zero (defection) always leads to a player being permanently excluded (because they become instantly stuck in a poverty trap – in essence, the proportional mechanism adopts a strategy akin to Grim-trigger[36]), whereas under an equal baseline, individual defections go unsanctioned (until the pool is exhausted). By contrast, the agent seemed to learn to temporarily exclude defecting recipients, typically

withholding offers for ~1-5 rounds but then making a more generous offer on the "re-inclusion" step, presumably to coax players back into the game (Fig. 2D). The mean of exclusion durations per game differed significantly between the RL agent and other baselines (*RL agent <proportional*, $z = 10.32$, $p < 0.001$; *RL agent <mixed*, $z = 7.87$, $p < 0.001$; *RL agent <equal*, $z = 11.54$, $p < 0.001$). The overall pattern is reminiscent of the successful 'tit-for-two-tats' or 'generous tit-for-tat' strategy in iterated prisoner's dilemma, whereby the agent is prone to punish but quick to forgive[37–39]. Another interesting way of visualising each mechanism is to plot the relationship between the offer on trial $t$ and the reciprocation that players offered over adjacent trials (lagged from $t - 4$ to $t + 4$ steps). For example, for the proportional mechanism, the offer is entirely given by the player's reciprocation on the previous trial (Fig. 2E, right panels), and the same is true to a more graded extent for mixed and equal baselines. However, for the RL agent, the peak is at $t = 0$, implying that the offer predicts the reciprocation rather than the other way around. Thus, the agent seems to learn to coax the player into reciprocating with generous offers rather than the player influencing the agent's behaviour.

Although we can scrutinise it in this way, the RL agent's policy remains hard to understand or explain. A more general contribution would be to distil these intuitions about the successful policy into a simpler explainable mechanism whose policy approximates that of our deep RL agent, but which could be explained to players. Based on our explorations of the agent policy, we thus devised an *interpolating* baseline that approximates the RL agent allocation mechanism, in which equality of allocation depends on the size of the common pool. This model was similar to other baselines, with the exception that the mixing parameter $w$ was allowed to vary with pool size. Using our virtual players, we tried out a family of interpolation functions that map from pool size to $w$ (via a function of the form $w = (R/200)^k$ where $\log(k) \in [-5, -4.9 \ldots 5]$) and picked that which maximises surplus for virtual players. Examining the outcomes of these interpolation baselines, we learned that the key feature for (predicted) success was that the pool had to be allowed to grow quite large before more egalitarian redistribution was permitted, because otherwise "reincluded" players received small or negligible allocations which did not allow them to properly participate in the economy (see Fig. S4 and Fig. S5 for more details). Thus, the most successful interpolating baseline had a high exponent, such that it was roughly proportional unless the pool was almost full (Fig. 3A).

To test the efficiency of this new baseline with real participants, we then ran a new experiment (Exp. 2) using three groups of human players: (1) a proportional baseline, and (2) the interpolating baseline, and (3) the deep RL agent. To maximise transparency, we provided participants with clear explanations about how the baseline mechanisms would behave (see Methods). Simulations involving our virtual players predicted that the interpolating mechanism would perform very well, and probably indistinguishably from our RL agent. This is indeed what we found (Fig. 3B). The RL agent and the interpolating baseline both generated higher surplus than the proportional baseline ($z = 2.9$, $p < 0.01$ and $z = 2.52$, $p < 0.01$ respectively) but did not differ from each other. Both mechanisms also maintained more active players than the proportional baseline ($z = 3.45$, $p = 0.001$ and $z = 2.88$, $p < 0.01$ respectively). The interpolating baseline in fact had lower Gini coefficient than both the proportional baseline ($z = 4.17$, $p < 0.001$) and the RL agent ($z = 3.67$, $p < 0.001$), indicating that it offered an excellent compromise between prosperity and equality. When we examined the Gini of the mechanism offer as a function of pool size, it was difficult to differentiate between the interpolating baseline and RL agent, as anticipated based on our virtual player simulations (Fig. 3C). Moreover, whilst the lagged regressions showed a subtly different pattern, the distribution of exclusion durations of the interpolating baseline show a similar left-skew as the RL agent (Fig. 3E). Once again, the virtual players allowed us to make accurate predictions about the outcomes

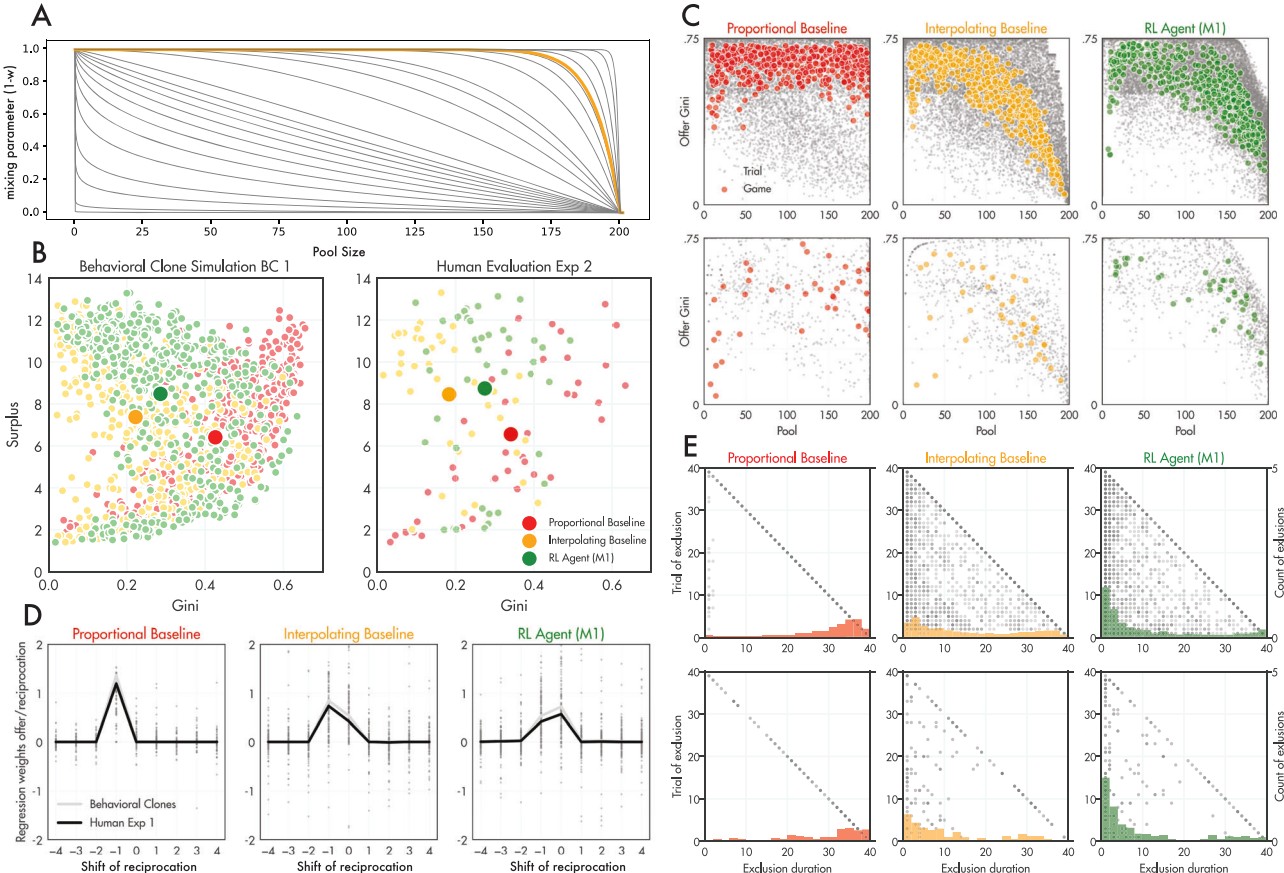

**Fig. 3 | Results of first trained mechanisms against novel, interpretable mechanism. A** The family of exponential functions that determine $w$ as a function of pool size (grey lines) and the one that produced the highest surplus with virtual players (yellow line; $(R/200)^{22}$). **B** The surplus and Gini coefficient generated from games played under two baseline mechanisms (proportional, red; interpolating, yellow) and the RL agent (green), for virtual players (left panel) and human participants (right panel) in Exp. 2. Each small dot is a game; the larger dot is the mean over games. **C** Same as Fig. 2C but for Proportional, Interpolating and RL agents. **D** Same as Fig. 2E. **E** Same as Fig. 2D.

with human players, even though they had not been trained on any data involving the interpolating baseline.

Next, we capitalised on the large and varied human dataset we had collected during these evaluations (comprising 453 additional groups) and used it to train new virtual players which, unlike our initial versions, had experienced high-performing mechanisms including the RL agent and interpolating baseline. We used these virtual players to train a new RL agent (M2) and pitted this new agent against our baselines in a final head-to-head Exp. 3. Because we anticipated that the RL agent and interpolating baseline would be well matched, we recruited double the number of participants (80 groups per mechanism). We found that indeed, this new RL agent did achieve higher surplus than the interpolating baseline ($z = 2.35$, $p < 0.05$) but that once again this came at the expense of equality, with the interpolating baseline generating an overall lower Gini coefficient for player surplus ($z = 5.64$, $p < 0.001$) as roughly predicted by the virtual players (Fig. 4A; see also Fig. S6). In further simulations, we explored this trade-off between surplus and Gini in more detail (Fig. S7) as well as the various mechanisms' ability to shape players towards the optimal reciprocation ratio of $1/(1+r)$ (Fig. S8).

We also explored the subjectively reported preferences of human players for each of the mechanisms we deployed in Exp. 2 and Exp 3. Surprisingly, although the RL agent generated large surplus overall, human players unequivocally preferred the interpolating agent. It was judged to be fairer, more understandable, and more prone to encourage cooperation, and players were clear that they would prefer to play again with this mechanism; see Fig. 4 (and Fig. S9 for results from Exp. 2).

One final concern is that our results may be due to the relative inexperience of participants with the mechanisms, which they experience over just 40 successive rounds of allocation and reciprocation. A related issue is whether the results we obtained may be due to the incentive structure we imposed, where all games lasted 40 rounds (but rewards only accrued from a subset of these). To address these potential issues, we conducted a new experiment (Exp 4, Fig. 4C), in which a new cohort of players ($n = 80$ groups of four) played the game with the M2 agent (or the proportional baseline with additional instructions) for three successive games in a row. Each game lasted a minimum of rounds and then ended with a probability of 0.2 after each additional round (leading to an average of $27.5 \pm 4.5$ trials per game). Participants who played with the mechanism M2 generated a larger surplus in game 1 ($z = 2.22$ $p = 0.026$), game 2 ($z = 4.99$ $p < 0.001$) and game 3 ($z = 6.08$ $p < 0.001$), relative to a proportional baseline with augmented instructions. Of note, the surplus obtained grew under M2 from game 1 to 3 ($z = 2.84$ $p = 0.005$), but decreased for the baseline proportional mechanism ($z = 2.41$ $p = 0.016$). In other words, as participants become better acquainted with the mechanism, it becomes more effective in promoting sustainable exchange.

## Discussion

This work makes three contributions. Firstly, we show that it is possible to accurately model the complex temporal dynamics of human multiplayer exchange using simple neural network models. The resulting simulation offered a remarkably accurate 'sandbox economy' that we could use to successfully predict the

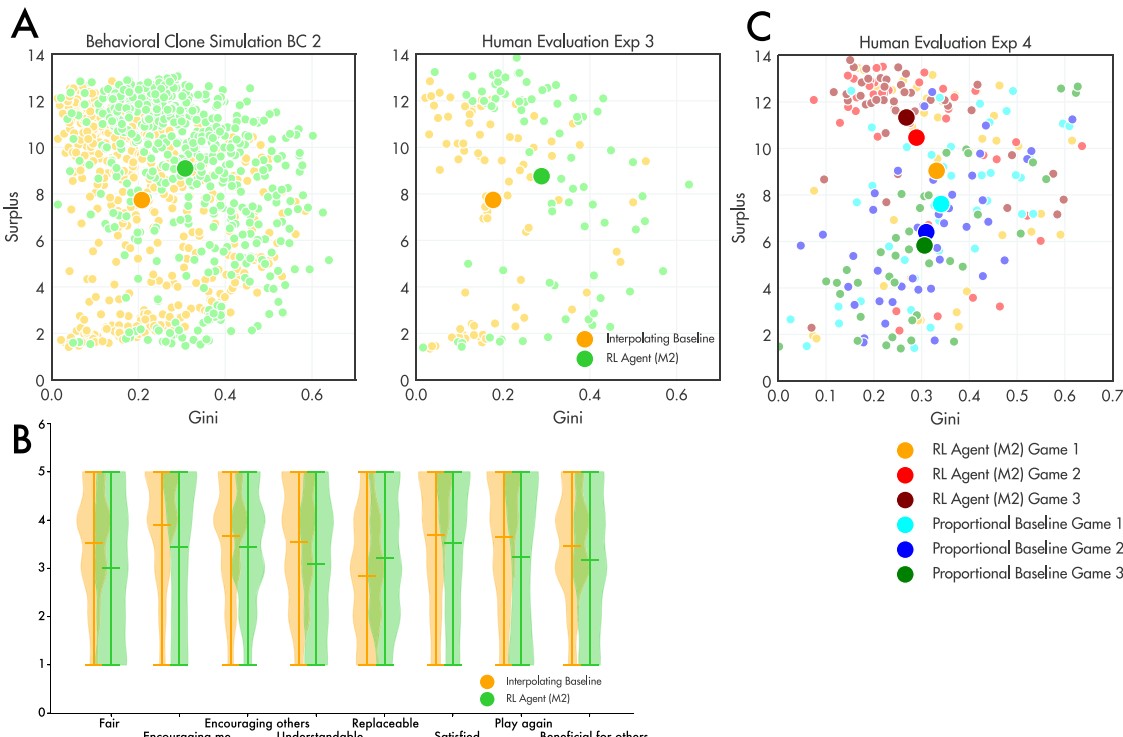

**Fig. 4 | Results of second trained mechanism. A** The surplus and Gini coefficient generated from games played under the interpolating baseline (yellow) and the RL agent M2 (light green), for virtual players (left panel) and human participants (right panel) in Exp. 3. Each small dot is a game; the larger dot is the mean over games. **B** Average reported preference for the interpolating (yellow) and RL agent M2 (green) agents on a number of dimensions. **C** Same as A; Results from human participants in Exp. 4., in which groups of players play three consecutive games.

consequences of various resource allocation schemes on surplus, equality and economic inclusion.

Secondly, using this sandbox, we show that deep RL mechanism can be used to discover a resource allocation policy which – when evaluated on new, unseen groups of human players – successfully to promote sustainable exchange, in terms of the highest levels of return to recipients (surplus), equity (Gini) and inclusion (proportion of included players). It achieves this even in the absence of endogenous mechanisms that allow players to communicate or self-organise, and in a qualitatively different way from baseline mechanisms based on unrestricted welfare (equal offers) or strictly conditional cooperation (proportional offers) or mixtures of the two. Unlike these baselines, the agent learns a mechanism that generates a positive relationship between prosperity and equality, so that games generating higher surplus are also characterised by a lower Gini coefficient. The success of this mechanism seems to hinge on four main factors. First, the mechanism has a generalised tendency to implement a more egalitarian allocation policy when resources are more abundant. Secondly, it tends to exclude free riders when resources are scarce. Thirdly, it is more likely to give to those who are prone to replenish the pool. Fourth, it offers brief, reversible penalties for defection that avoid creating poverty traps. This dynamic shifting of the mechanism between a more efficiency-focused versus egalitarian policy may offer a way to overcome previously documented, opposing preferences of elite policymakers and average citizens, respectively[40]. In a different game without a common pool, this mixture of progressive redistribution and sanction is also the recipe that encourages participants to vote for the mechanism[24].

Lastly, we show that a heuristic mechanism that is designed to mimic these features of the RL agent policy elicited similar levels of sustainable cooperation from humans. This demonstrates that the key feature of a successful allocation mechanism is indeed the way that equality of offer varies with the available resources (we chose to mimic

this mapping with a power law function, but we recognise that this is not the only viable function that could be used to successfully capture this relationship). This "interpolation" baseline achieved comparable (Exp.2) or only slightly lower (Exp.3) surplus, with higher equality and greater approval from human participants on a range of indicators. Thus, the mechanism was explainable to participants, and indeed human players judged it to be easier to understand. Like the RL agent, the interpolation agent was more prone to exclude defectors when resources are scant, so that the pool builds rapidly on the back of strongly reciprocating players. However, once the pool is replenished, the agent re-introduces excluded players, which has the effect of bringing average reciprocation close to optimal. This dynamic of inequality resembles that proposed to occur intrinsically during economic development at the level of nation states[35]. The fact that we can learn from RL models to build explainable, heuristic approximations opens a path for machine learning to help solve social and economic problems by informing, rather than replacing, human policymakers.

The machine learning architecture and pipeline used to tackle the problem is very general. The model of human behaviour makes no assumptions about the structure of the game, its inputs and outputs or what humans should aim to maximize. The mechanism, being equipped with the ability to retain memories within an episode, is similarly flexible. While we varied different factors of the game (e.g. Fig. S2 explores longer games), the pipeline itself should be amenable to yet more radical changes in the game structure (e.g. games more focused on risk or coordination) or input space (e.g. equipping the agents with convolutional neural networks should enable the processing of video games from pixels). An exciting challenge would be to find real world applications in which both initial data is available and in which exploring interventions via a mechanism is safe (e.g. designing auctions, recommendation algorithms, managing queues in an amusement park, setting incentives for contributions in virtual communities). The inclusion of memory to accommodate rich

dynamics within long episodes is a key technical advance over ref (ref. 23). Retaining activations over the whole episode makes the setup more compatible with a much more general class of games, rather than a narrow set of stylized strategic games.

In closing, we note some limitations of this work. Our participant cohort was drawn exclusively from the UK and USA. We do not know whether our results generalise beyond this demographic; moreover, we did not collect more fine-grained demographic data, so we do not know how whether our mechanism may work more effectively for (say) groups that differ by gender, age, or other relevant variables. Secondly, we note that the game employed here is somewhat different from the classic CPR problem, in that rather than drawing down freely from the common pool, players decide what to keep or reciprocate from a pre-set allocation, decided by the mechanism designer. This follows the logic of an iterated trust game (one way of thinking of this innovation is that the mechanism designer can set quotas on extraction). This formulation highlights the temporal dynamics within an episode in which players need to build a relationship over time, and motivates the implementation of a mechanism that can retain activations within an episode. Finally, whilst the mechanism discovered by deep RL is successful in the stylised game we employed here, this does not provide guarantees that similar principles would play out in more complex, naturalistic environments.

## Methods
### Materials and methods
#### Human task and interface
**Participants.** All participants were recruited from the crowdsourcing website Prolific Academic (https://prolific.co) and gave informed consent to participate. The study was approved by HuBREC (Human Behavioural Research Ethics Committee), which is a research ethics committee run within Google Deepmind but staffed/chaired by academics from outside the company. The final dataset contains 4952 participants. Participants joined groups of 4 via a lobby system and were allocated to games on a first come, first served basis. The entire experiment lasted -15–35 min. We collected no demographic data. No statistical method was used to predetermine sample size. No data were excluded from the analyses (except games with incomplete data, i.e. players dropping out).

**Game dynamics summary.** The game was a multi-player adaptation of an iterated trust-game with a persistent common pool resource. The resource pool was initialised at 200 units. The games lasted for 40 timesteps. Each timestep began with each of the four players receiving an endowment from the mechanism and deciding how much to reciprocate. Whatever amount was reciprocated was multiplied with 1.4 and added back to the pool. The pool had a maximum of 200 units and could not recover if it fell to zero. Whatever was not reciprocated was the surplus of the player on that timestep. The mechanism reward was the sum of all players' surplus over all 40 trials. In the cover story, participants were told the resource unit is 'flowers', in a field of flowers (the pool) that is controlled by a manager (mechanism).

**Instructions.** Participants (human players) began the task with instructions and a tutorial. The instructions informed them that they would be playing an "investment game" where they could earn "points" depending on both their "own decisions, and the decisions of others". Participants were instructed that they would "receive a base payment for completing the task as well as a bonus depending on the number of points they earned". Instructions contained screenshots of the interface participants would be using. Following these, participants played a tutorial round of the investment game lasting 3 timesteps. Data from the tutorial rounds were not included in the analysis and did not count towards participant bonus.

Participants did not know the precise number of time steps in each episode (although they did expect the experiment to last between 25 and 45 min in total). To discourage participants from planning with a specific time horizon in mind they were explicitly told that they would be paid in proportion to their surplus between trial 1 and a randomly chosen trial of the game. The implications of this randomised termination condition were further emphasised with the following sentence: "This means that you should continuously play as if the game could end at any time. Because of this, be aware that the further you are into the game, the less likely you are to actually be paid the money earned late in the game". Participants' total compensation averaged around 15 £ an hour.

**Interface.** The interface is shown in Fig. S12. The units in the resource pool were denoted in flowers and the mechanism called a "manager". Players viewed a table which showed the allocations from the mechanism to each player, including themselves. They then adjusted a slider to indicate which proportion of their endowment they wanted to reciprocate. The reciprocation decision is framed as being in units of "coins" (one flower turns into one coin). The fact that the reciprocation will grow by 40% to replenish the pool is highlighted in the interface. Players can increment their reciprocation in integers, which means that if they receive an endowment of below 1 they are obliged to pay themselves the entire amount and reciprocate 0. Players have 90 seconds to respond and are replaced by a uniformly random responding bot if they time out twice (on the first timeout their response was recorded to whatever the contribution slider was set). Note that games in which any player dropped out were not used for training or analysis (this affected about 10% of games). Each player saw themselves displayed as 'You' (i.e. player 1).

At the end of each round, an overview screen appears that summarises the change in the pool, the offers made, and each player's reciprocation. The screen also displays the running total number of coins that each player did not reciprocate so far in the game. In addition, the overview screen shows the total bonus earned by the players in pounds (£), with a conversion ratio of 1 game point to 0.008 £.

For experiments 2, 3 and 4, instructions also gave players a written explanation of the strategy of the manager (i.e., the mechanism that was controlling the resource pool).

For the Proportional Baseline the explanation read: "The manager will offer flowers proportional to the last re-investment. For example, if half of the total reinvestment last round was done by you, you get half of the flowers this round. If a player is the only one who re-invested, they will get all the flowers next round. Generally, the more you re-invest the more you get offered, but it is always relative to other players."

For the Interpolating Baseline the explanation read: "The manager will adjust its policy to the flower field size. When there are a lot of flowers in the field, the manager will tend to give flowers to everyone, no matter how much or little they reinvested. If there are few flowers in the field, the manager will tend to give more flowers to those players who re-invested the most on the last trial (relative to other players.)"

For the RL agents the message was intentionally less informative and just focused on the goal of the manager: "The manager you will play with, has the following strategy: The manager aims to offer flowers in such a way that all players make the maximum amount of money possible over the course of the experiment."

In experiments 2 and 3, participants were asked to complete a questionnaire to indicate their attitude towards the manager they had just played with (see below).

In experiment 4 the game setup was altered in 3 ways.

First, players played 3 games in a row (staying within the same group). They were always informed when a new game started and the pool reset to the maximum.

Second, the implementation and instructions for how long games were simplified. Players were told that each game has a minimum length of 25 rounds and after that an 20% chance to end the game. Players were paid for the whole duration of the episode, but the episodes had variable lengths.

Third, for the proportional baseline players got an expanded instruction that clearly explains what behaviour is sustainable in the task: "You can choose how much to keep from the offer and how much to re-invest from the offer to the flower field in order to sustain it. If everyone keeps 29% of each offer, then the flower field can be sustained indefinitely (because the re-investment grows). However, if one player takes >29%, this player may make more money than the others. However, if all of the players take >29%, then the flower field will shrink. In short, each player individually can be better off taking >29%, but for the flower field to be sustained the group as a whole has to act sustainably."

## Mechanisms

The mechanisms that managed the resource pool were either deep reinforcement learning agents or simple baselines. Details of how they were constructed are described below.

To generate the initial training data (prior to experiment 1), we collected data in which humans played under two baselines. The first baseline allocated the pool randomly between players. The second baseline was defined by the mixing parameter $w$ (see Eq.1), which was sampled randomly on each game, providing a spread from equal to proportional behaviour. The baselines left a residual fraction of funds in the pool, determined by an additional parameter. This data (303 games) was then used to train an initial piloting agent M0 with which 234 games were collected. This data collection exercise (537 games of four players total) was otherwise identical to that reported above. The resulting data were used for imitation learning, to generate the virtual players used in both experiment 1 and experiment 2.

## Human datasets

Below, we provided details of the datasets that are reported in the main text. In all cases, we continued data collection until we had at least 40 games per group (80 in experiment 3). We then excluded excess games to ensure balanced numbers of groups across conditions.

**Train Set experiment 1.** This comprised 537 games of both random and baseline mechanisms, as well as an early prototype of the RL agent (see above). Used to create BCs BC1 that were used to create Mechanism M1 (and M1' without memory). These were evaluated in experiment 1 and 2.

**Eval Set experiment 1.** This comprised 40 games each of 3 baselines (Equal, Mixed and Proportional) and 2 Agents (M1 and M1'). We collected 213 games in total. Note that the 13 excess games are due to the fact that during data collection the mechanism is randomly allocated to participants, and therefore data collection per condition can overshoot the target. In this case we only analyse the first 40 games gathered in each condition but use all collected data for later training.

**Eval set experiment 2.** This comprised 40 games each of 2 Baselines (Proportional, Interpolating Power 22) and the agent M1. Here people were instructed about what the mechanism aims to achieve and its strategy. We collected 143 games in total.

**Train set experiment 3.** This comprised all aforementioned data, plus some additional exploratory evaluations we did before experiment 1 (that initially discovered the strength of M1) for a total of 990 games. These were used to Create the BC2 group that then were used to train Mechanism M2.

**Eval set experiment 3.** This comprised 80 games each of Mechanism M2 and Interpolating Power 22 baseline (with instructions). We collected 163 games in total.

**Eval set experiment 4.** This comprised 40 sets of 3 games (played by the same players consecutively) each of Mechanism M2 and Proportional baseline (with expanded instructions).

This data is available at https://github.com/deepmind/sustainable_behavior/.

## Data analysis and figures

For Fig. 2A, we plot (1) the mean of the aggregate surplus and (2) the coefficient of the aggregate surplus obtained in each game (data points represent games). The surplus of each game is aggregated across players and trials (i.e., the amount they were offered and did not recontribute). The Gini coefficient is calculated on the aggregate surplus each player has achieved across all trials. We use the Wilcoxon rank-sum tests to compare the surplus and Gini coefficient values achieved in each experiment (with group being the unit of replication).

Fig. 2B overlays dots (games with humans) over contours (games generated by BCs displayed via a kernel density estimate with a threshold of 0.01, 8 levels for the distribution (depleted or sustained games) with more data points, and the number of levels for the smaller distribution scaled proportionally). We distinguish games in which the pool was sustained (>1 on the last trial) or depleted. We analyze the surplus and Gini coefficient, alongside 'trial of pool depletion' and 'active players' in the same way. The trial of pool depletion is the trial in which the pool drops below 1. The *active players* variable is the mean over trials of how many players received an offer of 1 or greater in each round. We also calculate Pearson correlations between surplus and Gini coefficient, separately for sustained and depleted games.

Fig. 2C plots for each trial (and means across trials per game) the pool size at that trial and the Gini coefficient calculated over the offer the mechanism gave to players on that trial.

Fig. 2D plots trials in which a player that on the previous timestep received an endowment of 1 or higher, got an offer of <1, i.e., an 'exclusion'. These events are plotted on the axes of which trial in the exclusion occurred and how long the exclusion lasted. Dots on the diagonal indicate exclusions that last to the end of the episode (trial of exclusion and length of exclusion add up to the length of the episode).

Fig. 2E shows regression weights (trials are data points) and their median (over all trials) calculated per trial, describing the relationship of the offer made by mechanism in dependence to the players reciprocation (considering past and future behaviour of the players, minus to plus four trials). For example, for the proportional baseline the only weight of notable size is the -1 weight, which indicates that the current offer is proportional to the reciprocation on the previous trial (which is exactly the behaviour this mechanism implements).

All plots and statistics were reproduced for experiment 1, 2 and 3 in the same way. Note that python code that produces the plots and statistics is available at https://github.com/google-deepmind/sustainable_behavior/.

## Questionnaires

At the end of the experiment, participants were asked to rate their level of agreement with a series of 8 statements (in the order below) on a 5-point scale.

1. The manager's policy was fair.
2. The manager's policy encouraged ME to contribute.
3. The manager's policy encouraged OTHERS to contribute.
4. The manager's policy was easy to understand.
5. I can think of a policy that would have been better for everyone.
6. I am satisfied with the money I made from the game.
7. If I played again I would like to play with this manager again.

8. This manager encouraged me to contribute in a way that was beneficial to others.

We compare the average (across participants) the agreement ratings with a rank-sum test. To control for multiple comparisons, we submitted all *p*-values (per experiment) to FDR correction.

## Behavioural cloning datasets

For analysis of the *behavioural clones*, we unroll 40 or 512 episodes of BC1. Note that BC1 is an ensemble of 4 BCs that were sampled with replacement during training (see section "Training virtual players (BC1)" below), but for analysis or evaluation purposes we do not sample them. Instead, each of the 4 BCs is in a fixed slot and all episodes contain all 4 BCs. Note that for experiment 3 we unroll 4 BC2s.

## Training pipeline for the RL Agent (M1)

Our training pipeline for the RL Agent (M1) consisted of five main steps: (1) developing initial baseline mechanisms based on the economic literature; (2) collecting data from human players playing under the baseline mechanisms identified; (3) training virtual human players using supervised learning to imitate the recorded human trajectories; (4) training an agent with a deep RL method to maximise cumulative surplus when interacting with virtual players; and (5) evaluating the RL agent by deploying it with new human participants, along with comparison baselines.

**Baseline mechanisms.** As outlined in the main text, our initial baseline mechanisms were hand coded. The offers of these mechanisms were determined at each round *t* by taking a weighted sum of the proportional and equal allocation policies, with the weight controlled by a mixing parameter *w*. This was calculated according to the formula in Eq. 1.

For initial data collection only, we also varied another parameter which controlled the fraction of the pool that remained unallocated (varying from leaving 0% to 40% of the pool unallocated). However, we observed that the highest-performing mechanisms always allocated all of the pool, so in subsequent data collection, we dropped this parameter.

As part of the initial set of baselines, we also included a random mechanism that drew at each round *t* five random proportions from a Dirichlet distribution with concentration 1. These proportions were multiplied by the current pool size to obtain the random offers to be made to each player and the amount to be kept in the pool.

**Initial data collection.** As mentioned above, our data collection process involved human participants playing under various mechanisms, during which we record all observations and actions taken by players when exposed to the game's dynamics. This included the state of the pool, mechanism offers, and the actions and observations of other players. Our initial dataset, Train Set 1, was produced during the first phase of data collection and includes 537 games. Among these games, 36 were generated with participants playing with a random mechanism, 303 were played using hand-coded baselines with varying mixing parameters *w*, and 234 were played under a prototype neural mechanism (M0).

**Training virtual players (BC1).** Based on **Train Set 1**, we trained virtual human players via behaviour cloning[41], an imitation learning technique that involves training virtual players or "clones" to mimic human gameplay.

A virtual human player is a deep neural network that emulates the behaviour of a single player. This means that we use the observations of a single participant to predict their action at each timestep (since we have four players, we obtain four times as many reciprocation actions as rounds in an episode). To account for the inherent noise in human

data, we employed a probabilistic neural network to model the behaviour of the player. Specifically, we model their action predictions as a categorical uniform distribution, which is explained in further detail below.

The neural network's inputs match the observations of the real human players. The network takes in: the offers made by the mechanism to each player in the current round, $e_{i,t}$ (represented by four real numbers), the contributions made by all players in the previous round $c_{i,t-1}$ (other four real numbers), and the current size of the pool $R_t$ (one real number), resulting in a 9-dimensional input. All inputs were normalised by being divided by 200, which denotes the maximum size of the common pool. The output of the network is a single number representing the prediction of the focal player's contribution for the next round, $\hat{c}_{i,t}$. The virtual player network architecture comprises a memory core, namely Gated Recurrent Units (GRU)[42], surrounded by several fully connected (FC) layers (see Table S1 for details). The first 2 fully connected layers encode the input into an initial neural representation, which is then passed to the memory layer. The fact that recurrent neural networks are part of the architecture of the virtual players means that they can potentially learn to keep track of the past and use the game history to make their predictions about what to contribute this round. The final 2 FC layers make a set of non-linear projections, followed by a final linear projection onto a N-dimensional space, representing N bins of a categorical distribution. These bins split the space of continuous numbers form 0 to the offer $e_{i,t}$ received by the focal player this round. A softmax function applied to these N logits to discretely determine the most likely bin, call it max_bin = argmax(softmax(logits)). Then, we sample uniformly from the continuous interval [max_bin, max_bin+1) to determine the proportion of the offer received that the focal player would reciprocate this round, $\hat{c}_{i,t}$. Please refer to Fig. S15, which illustrates a schematic representation of the architecture employed for modelling virtual players.

The virtual player network was trained using back-propagation through time to minimise the cross-entropy loss between predicted and actual contributions. During training, we used mini batches of size 256 and employed Adam optimization with an annealing learning rate that started at 0.0005 and decayed exponentially by 0.05 every 1000 steps until reaching 0.000005. No regularisation was applied. The model was trained for 700,000 update steps, and we selected four checkpoints with high surplus in simulations with hand-coded baseline mechanisms and low surplus when playing with the random baseline mechanism. The ensemble of these four BCs is what we call the *BC1* group.

All hyper-parameters and architectural details are presented in Table S1. In Fig. S11, we show side-by-side comparisons of human and behavioural cloned data.

**Training the RL agent (M1).** Our aim was to create an allocation mechanism that could distribute resources in a way that maximised surplus. To achieve this, we designed a virtual environment with the game dynamics presented in the main paper. We replaced the human participants with four virtual players that we had trained using the methods described above. Using deep reinforcement learning (RL) and by letting the agent interact with virtual players, we trained the mechanism (or RL agent) to maximise the aggregate player surplus. After convergence, we use it to play games with humans (or virtual players) for a custom number of rounds.

We modelled the RL agent as a deep neural network and employed an architecture based on Graph Neural Networks (GNNs)[43]. Using GNNs ensured two desirable properties under our game design: (1) a uniform opening move, meaning that the mechanism would make equal offers on the first timestep, and (2) equivariance to permutation in the ordering of participants.

The RL agent took in a 9-dimensional input, consisting of the agent's endowments from the previous round (4 real numbers), the

contributions it received from the other players in the previous round (another 4 real numbers), and the current size of the pool (1 number). To ensure consistency across inputs, all values were normalised by dividing them by the maximum attainable value (which is 200). The mechanism network produced a 5-dimensional output, which was passed through a softmax function to ensure that the values were positive and summed to 1. These values, when multiplied by the pool size, determined the endowments to be offered to each player in the next round and the amount of the pool that should be retained.

The network architecture of the RL agent was based on GNNs. We arranged the observations into a fully connected directed graph $(u, V, E)$ where each player was represented as a vertex $v_k \in V$ with three attributes: its past endowment, its past contribution, and the current pool, all normalised. Directed edges $e_{sr}$ connecting $v_s$ and $v_r$ had empty initial attributes, and the input global attribute vector $u$ was filled in by the pool. Computations in Graph Networks start by updating the edge attributes, followed by the node attributes and finally global attributes. In particular, directed edge attributes were updated with a function $\varphi_e$ of the input edge attribute, the sender and receiver vertex attributes, and the global attributes vector: $e_{s,r'} = \varphi_e(e_{sr}, v_s, v_r, u)$; vertex attributes were updated as a function $\varphi_v$ of the input vertex attributes, the sum of all updated edge attributes that connect into $v_r$ and the global attributes vector: $v'_r = \varphi_v(\sum_s e'_{s,r}, v_r, u)$; finally, the global attributes vector was updated with a function of the input global attributes, and the sum of all updated edges and vertices: $u_{r'} = \varphi_u(\sum_{s,r} e_{s,r'} \sum_k v'_k, u)$. We note that the same functions $\varphi_e$, $\varphi_v$ are used to update all edges and nodes in a graph, and that both the input and output of Graph Networks are directed graphs, so these modules can be used in sequence.

The mechanism's policy network consisted of two sequential GNNs. In the first GNN, we implemented all of $\varphi_e$, $\varphi_v$ and $\varphi_u$ as distinct non-linear fully connected layers with 32 output sizes and ReLU activation functions. The output of this GNN was then passed to the second GNN, where we implemented $\varphi_e$ as a non-linear fully connected layer with 32 output size; $\varphi_v$ as a memory layer (Gated Recurrent Units) with 16 hidden sizes, followed by a non-linear layer with 32 output size, finally followed by a linear layer with a single output unit; and $\varphi_u$ as a non-linear fully connected layer with 32, followed by a linear layer with a single output unit. We then concatenated the node outputs (one per player) with the global output (one) and normalised the concatenation (total size of 5) using a softmax function. This yielded the redistribution weights, which sum to 1. When multiplied by the current size of the pool, they determine the absolute values of the endowments to be offered to each player, and the amount to be kept in the pool. Note that the model is deterministic, so that given the same inputs, it will always produce the same outputs.

The diagram below illustrates the architecture of the network used to model virtual players with behaviour cloning.

We trained the described policy network of this mechanism gradient descent to maximise the cumulative player surplus. Specifically, the training objective was $\max \sum_{t=0}^{T=40} \sum_{i=1}^{N=4} (e_{i,t} - c_{i,t})$.

During the RL agent training, we only used virtual players to roll out episodes. No human data was utilised to predict true player contributions, meaning that there was no teacher forcing involved in the mechanism training process. The mechanism was trained on the BC1 group, which is composed of the four virtual players trained during the previous step. We sampled from BC1 with replacement at the beginning of each episode. Note that the offers seen by the virtual players were generated by the mechanism policies, which may lie outside of the human data, thus requiring the virtual players to generalise beyond the training dataset.

To optimise our model, we used Adam optimization with an annealing learning rate starting at 0.001 and decaying by 0.05 every 1000 steps until it reached 0.00001. We trained our model using mini batches of size 256 and did not apply any regularisation. The model

was trained for 500,000 update steps, and we saved a checkpoint every 50,000 steps. We evaluated the performance of the frozen models in simulations with virtual players and selected the checkpoint with the highest surplus as our final model, which we refer to as *M1*. All hyper-parameters and architectural details are presented in Table S2.

We also trained another RL Agent via the same process as M1 but with a slight difference in its network architecture. Specifically, its second GNN was identical to the first one, lacking Gated Recurrent Units in the node computations and using only fully connected layers (i.e. it is purely feedforward and has no recurrency). We refer to this agent as M1'. In order to save space, we do not report data from this agent in the main text (it performed similarly to M1 but was slightly less effective).

Prior to the development of M1, we created a set of prototypical RL agents referred to as M0. During our initial pilot human data experiments, some participants played under the M0 mechanisms. The data collected using M0, along with the data collected using the baseline mechanisms, was incorporated into the Train Set 1. As a result, the trained set of BC1 includes trajectories obtained under these preliminary agents, M0. The prototypical mechanisms had simple network architectures consisting of either fully connected layers only or fully connected layers coupled with a lightweight GRU or LSTM memory core. At this stage, we did not engage in hyper-parameter finetuning or strategic performance selection. As previously mentioned, we later discovered that a GNN-based architecture was crucial for a strong mechanism policy. Note that the prototypical architectures of M0 did not include any GNNs.

### Evaluation
Our evaluation method involves deploying the RL agent trained in our experiments with new human participants, alongside comparison baselines. In Exp. 1, the comparison baselines consist of three variations of the weighted baseline described in the main text: the Proportional Baseline ($w = 0$), the Equal Baseline ($w = 1$), and the Mixed Baseline ($w = 0.5$). For more details, see section on Eval Set Exp 1.

For subsequent experiments, we draw on the insights gained from our in-depth analysis of the previously trained mechanism, RL Agent (M1), and introduce an additional baseline, the Interpolating Baseline. The offers of this baseline are determined by a power-law function of the normalised pool size, namely $(R/200)^k$, where we empirically determined that $k = 22$ is the power coefficient that maximises player surplus when interacting with virtual players. For more details, see section on Eval Set Exp 2.

### Training pipeline for RL Agent (M2)
The final step of the RL Agent (M1), which involved evaluating it under various baselines, yielded new data that allowed us to iteratively refine both the virtual players and the mechanism by repeating the steps outlined above. Please refer to the Figs. S14 and S17 for an illustration of this process.

### Training virtual players (BC2)
We used the same network architecture and training procedure as for BC1, but made some adjustments to account for the increased dataset size. Specifically, we employed larger networks, larger batch sizes, and a lower learning rate (see Table S1 for details). We selected the top 8 virtual players from differently seeded runs and at different checkpoints to form what we call the *BC2* group.

### Training the RL agent (M2)
We kept the same network architecture and training process as in M1, except for two adjustments. First, we modified the memory unit size to accommodate a wider range of potential human behaviours. Second, during mechanism training, we adjusted the number of behaviour clones to sample from during group formation to four, drawn from the BC2 group of eight virtual players trained in the previous step (see

Table S2 for details). These changes were necessary to accommodate the wider range of potential human behaviours and to effectively capture the increased variability and richness of the data. We selected the mechanism with the highest surplus under virtual players and refer to it as *M2*.

### Evaluation with new human participants
Once we completed the training of M2, we proceeded to evaluate its performance using a fresh group of human participants. Further details can be found in the Eval Set Exp 3 section.

### Implementation details
Our entire codebase was implemented in JAX. The policy distribution of the BC was implemented using distrax CategoricalUniform which is open-sourced, and that allows computing the gradient of the distribution parameters from the samples (publicly available at https://github.com/deepmind/distrax/blob/master/distrax/_src/distributions/categorical_uniform.py). We utilised the Jraph library (https://github.com/deepmind/jraph) to implement the graph neural networks, and the Haiku library for all other neural networks. We leveraged pandas, matplotlib, and seaborn for data analysis and visualisation. We saved checkpoints of the behaviour clones and mechanism parameters approximately every 50,000 training steps. We typically noticed that over-parameterizing our networks and training them for longer resulted in improved performance (similar to the double-descent effect[44]). All our experiments were conducted on a single NVIDIA Tesla P100 GPU Accelerator and completed within 24 h.

### Reporting summary
Further information on research design is available in the Nature Portfolio Reporting Summary linked to this article.

## Data availability
This data is available at https://github.com/google-deepmind/sustainable_behavior/.

## Code availability
Code to reproduce all figures and statistics is available at https://github.com/google-deepmind/sustainable_behavior/.

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

## Acknowledgements
We thank David C. Parkes for comments on an earlier version of this paper and helpful suggestions. The authors received no specific funding for this work.

## Author contributions
R.K., M.P., A.T., J.B., L.L., R.E., O.P.H. and C.S. contributed to conceptualization, formal analysis, investigation. K.T., M.B. and C.S. supervised the project. All authors contributed to the writing of the paper.

## Competing interests
The authors declare no competing interests.
