## [Transparent Peer Review file · Nature Communications]

Deep reinforcement learning can promote sustainable human behaviour in a common-pool resource problem

Corresponding Author: Dr Raphael Koster

Version 0:

Reviewer comments:

Reviewer #1

(Remarks to the Author)

This MS mainly explores how to address social dilemmas by designing an allocation mechanism that can incentivize participants to make sustainable contributions to the public resource pool in situations where resources are limited. The authors first trained neural networks to simulate the behavior of human players, and then used deep reinforcement learning (RL) to train a social planner to maximize the total player return. This social planner has discovered a redistribution strategy that leads to a large surplus and inclusive economy, where players receive roughly equal benefits.

The main innovation of this study lies in the use of new tools from AI research, such as deep neural networks and deep reinforcement learning, to design and optimize resource allocation mechanisms. This method not only handles complex historical event sequences, but also maximizes social welfare within the group.

Overall, the contribution of this study lies in proposing a new perspective and method to address the social dilemma of resource allocation, which has positive implications for areas such as public interest provision, compensation, and welfare theory. The authors have successfully applied AI research tools to social science issues, which is an interdisciplinary attempt and opens up new research paths.

Although the MS is valuable, there are still a series of issues that need to be further addressed.

- 1) On page 3, line 135, $\Delta R_t = \sum_i (1+r)^e \sum_j c_j$, this formula seems to be problematic. It seems that 'e' should be before the first '+' sign.
- 2) An accurate comprehension of different roles and interactions in your model is crucial for the understanding of the readers. In Figure 1A, there is an instance (Right side) where the second player receives resources from the common pool but does not reciprocate back into the resource pool, leaving a remainder of zero. Here, it is suggested that the authors explain the basic principles behind zero remainder.
- 3) The authors have embarked on their research utilizing a repeated multi-person trust game, an approach that diverges from previous studies. Notably, the role of the investor is not played by a specific individual participant but is instead substituted with a resource pool (Similar in form to the public goods game). Intriguingly, this resource pool exhibits the capability to make intelligent decisions, determining the quantity of resources (depicted as blue flowers in the manuscript) to allocate to the game participants, or alternatively, to withhold them entirely. This innovative experimental design is compelling. Would the authors be able to elucidate the underlying motivation for adopting such a methodology? In addition, the role of a resource pool is similar to that of a conditional investor in a multiplayer trust game, who can choose allocation ratios based on the game environment. Authors should explain this.
- 4) The authors recruited 640 participants. I'm wondering whether the inherent attributes of these participants (such as gender ratio, education level, etc.) might have an impact on the experimental results. I don't seem to find related research or description on this aspect in the supplementary materials. Another point is that the supplementary materials state that the final database contains 4952 participants, which needs clarification.
- 5) While the MS suggests implementing a more egalitarian distribution policy when resources are abundant, the discussion

regarding how to adjust strategies to form the most optimal resolution in situations of extremely scarce resources is insufficient.

- 6) It would be beneficial to discuss the impact of several social factors circled around policy-making, such as social norms and demographic information, and how they might contribute to forming the optimal resource allocation strategy.
- 7) Despite the conducted evaluation with human players, it would perhaps be enlightening to delve deeper into demographics distinguished by varying backgrounds, cultures, and age groups (as mentioned in (4)). Further research on these specific populations might elucidate some novel and meaningful conclusions.
- 8) A series of experiments are designed to investigate the impact of different mechanisms on resource management through virtual environments, behavioural cloning and the training process of deep learning agents. The authors choose specific network architectures and hyperparameters such as learning rate and batch size. Can you elaborate on the reasons for choosing these parameters and their impact on the experimental results? In addition, can more specific information, such as the number of layers of the neural network, the number of neurons per layer, the choice of activation function, etc., be provided so that the reader can fully understand how the model works and what the effects are?
- 9) In the discussion section, the authors can suggest next steps for research on how deep reinforcement learning can be further applied to promote sustainable behaviours and how the limitations in the current research can be addressed. Although, deep RL mechanisms such as the one in the paper can be used to discover a resource allocation policy and are found to be easier to understand by the human participants, will humans ultimately be able to do the same thing as these AIs?

There are a few minor suggestions for improvement:

- 1) On page 3, line 134, *min* should be in straight font instead of italics.
- 2) In the caption for Figure 1, it is suggested that the authors provide a more detailed explanation of what 'w' represents. Although there is a relevant explanation in line 175, it should be explained in the caption of the figure.
- 3) Page 5, I recommend that the authors provide a more comprehensive explanation regarding the variable 'j' in the formula.
- 4) I recommend that the authors consider improving the resolution or clarity of Figure 2.
- 5) For the sake of uniformity, it is advised that the terms 'equal', 'proportional', and 'mixed' in lines 195 and 198 are presented in italics.
- 6) Line 291, the phrase "This behaviour is reminiscent of Kuznets theory" might be clearer if it were "This behaviour is reminiscent of the Kuznets curve theory".
- 7) The references on repeated multiplayer trust games, deep reinforcement learning, and public resource management need to be moderately increased.

(Remarks on code availability)
No code provided.

Reviewer #2

(Remarks to the Author)

Comments of "Using deep reinforcement learning to promote sustainable human behaviour on a common pool resource problem"

This paper provides an in-depth experimental validation for promoting sustainable human behaviour in a common pool resource game. It starts by using data collected from human participants to create clones that imitate human behavior. These clones are then used to train a reinforcement learning (RL) agent, M1. To test how generalizability the RL agent allocation mechanism is, the authors conduct Experiment 1, and find that the RL agent generated a higher surplus compared to other baselines. They investigate the properties of the RL agent further. To understand the reasons behind the effectiveness of the RL agent, the authors introduce a new baseline named the interpolating baseline and carry out Experiment 2. The findings indicate that this new baseline is as effective as the RL agent. This suggests that the success of the RL agent is due to its allocation mechanism being dependent on the size of the pool. Additionally, both human players and virtual players participate in the first two experiments. Their similar behaviors demonstrate the feasibility of using virtual players to train the RL agent. In addition, the authors made use of the data from the first two experiments to retrain new virtual players and a new RL agent, M2. Then they conduct Experiment 3 to revalidate that the new RL agent allocation mechanism excelled in maximizing surplus. Finally, to explore whether the previous outcomes are robust to the incentive structure, the authors altered the way of calculating bonuses in Experiment 4.

As a general assessment, this paper presents a very interesting conclusion. The allocation mechanism implemented by the RL agent promotes reciprocation among players more effectively than common allocation mechanism such as fair allocation

or allocation proportional to contribution. Additionally, it identifies why the RL agent allocation mechanism is effective in maximizing surplus: the mechanism depends on the size of the pool. Specifically, when the pool size is small, the allocation mechanism is close to a proportional baseline, whereas with a larger pool size, the RL agent tends to distribute more equally. This characteristic allows the agent to exclude free-riding players when resources are scarce, leaving behind players who reciprocate more. As resources increase, the allocation mechanism can then reinclude previously excluded players. Ultimately, this maximizes the total surplus while ensuring that the distribution of surpluses among players is not too unequal.

I have two main concerns about this paper.

(1) From my perspective, using a repeated multiplayer trust game framework to study the CPR problem may be inappropriate because both the objective and the equilibrium structure of the trust game differ significantly from those of the typical CPR problem.

In a traditional CPR scenario, players extract resources from a common pool without any obligation to reciprocate. The goal is to effectively manage and regulate the use of these resources to prevent depletion. However, in the described repeated multiplayer trust game, players receive endowments and decide how much they return to a common pool. This game focuses not on preventing the pool's premature depletion but on maximizing the total surplus for the players. Additionally, this game setup is more akin to a public goods game, where players receive an endowment and choose how much to contribute back. Contributions from players cannot exceed their initial endowments, and the benefits from public goods can be considered as endowments in the next round.

(2) This article explores the design of an allocation mechanism using AI tools. The method involves several key steps. First, data is collected from human participants to train virtual players. These virtual players are then used to train a reinforcement learning (RL) agent. Behavioral experiments follow, comparing the RL agent with baseline models. The RL agent's mechanism effectively maximizes total surplus, leading the authors to examine its properties further. This method is based on the approach used by Koster et al., 2022, as cited in reference [21]. Compared to the study by Koster et al., 2022, the main innovation of this paper lies in devising an interpolating baseline that approximates the RL agent's allocation mechanism. This further confirms that the success of the RL agent's allocation mechanism relies on the allocation being dependent on the pool size. However, the level of innovation in this paper (i.e., devising an interpolating baseline that approximates the RL agent's allocation mechanism) may not be sufficient for Nature Communications.

Small questions:

(1) In the interpolating baseline, w is defined as a power-law function of pool size. Given that the RL agent was more punitive when the pool was low, similar to the proportional baseline, and made more generous offers as the common pool grew, distributing resources more equally—why wasn't a piecewise linear function considered? If w is set as a piecewise linear function dependent on pool size, with a threshold where $w=0$ below this threshold and is linearly proportional to the pool size above it, could similar results to the RL agent be achieved?

(2) Both Experiment 2 and Experiment 3 indicated that the interpolating baseline generated a higher surplus than the proportional baseline. Given these findings, I'm curious why Experiment 4 chose to compare RL agent M2 with the proportional baseline instead of the interpolating baseline. Like in Experiment 3, it should be compared with the interpolating baseline to test whether the results they obtained may be due to the incentive structure.

(3) In the first paragraph of the 'Results' section, there is a subscript missing from "e" in the expression " $\sum_i e_{i,t} \leq R_t$ ". The expression for ΔR_t is incorrect. In " $\sum_i \sum_t s_t$ ", "s" is also missing a subscript. Additionally, both this section and the caption for Figure 1 mention " $r=1.4$ ", but it should correctly be " $r=0.4$ ".

(4) In the third paragraph on page 7, there is an error where Fig. 2B is mistakenly referred to as Fig. 2A. Similarly, on page 9, Fig. 4A is incorrectly cited as Fig. 3A.

(5) Could you provide a more detailed description of the superimposed colored histograms in Fig. 2D? My confusion arises from the observation that there are fewer bars than 40, which does not seem to correspond directly to the values on the horizontal axis. How are these histograms constructed, and what does each bar represent in relation to the axis values?

(6) The term "game" is frequently used in the text to refer to all the rounds played by a group, but the definition of "game" is not clearly stated, which leads to confusion about the scope of what "game" refers to. It would be better to include a clear definition of 'game' in the manuscript.

(7) On page 17, screenshots of the experiment show that each participant is assigned a number. Is this numbering fixed? For example, does the player labeled as Number 3 remain as Number 3 throughout the entire game on a specific participant's interface?

(8) On page 19, the 'Mechanisms' section mentions that the data used to train virtual players is collected under two baselines. However, this description conflicts with the information on page 22 in section 7.2 "Initial data collection", where it is stated that the training set also includes play under a prototype neural mechanism M0. I would suggest that the authors to make the descriptions consistent.

(9) In the first line on page 20, the text in parentheses following "2 Baselines" should not include "M1".

(Remarks on code availability)

Reviewer #3

(Remarks to the Author)

The authors have innovatively applied artificial intelligence as a tool to address the social dilemma of resource allocation. Specifically, they have utilized deep reinforcement learning (RL) to design a resource allocation mechanism in a multiplayer trust game, discussing how this mechanism can incentivize participants to sustain the sustainability of a shared pool. This represents a natural extension of applying AI research tools to studies on resource allocation. By comparing this mechanism with other allocation mechanisms, the authors have found its effectiveness in achieving collective benefits while also considering distribution equality. Through the demonstration of heuristic mechanisms, they have revealed typical allocation characteristics of RL mechanisms, showing a tendency towards more equal distribution when resources are more abundant. Although most results are obtained through comprehensive simulations, the motivation is clear, and the study holds significant value for research into cooperative behavior across various disciplines. Yet, we believe some improvement can be made.

- 1) Throughout the entire paper, the main focus and results revolve mainly around the allocation mechanism, but the description of participants' strategic behavior could possibly be more detailed. Particularly, while the authors consider the outcomes of virtual participants in the game, the strategic characteristics exhibited by these participants in the game are still unclear. In repeated games, there can be various types of strategies involved (See "Repeated prisoner's dilemma games in multi-player structured populations with crosstalk"; "Evolution of cooperation with nonlinear environment feedback in repeated public goods game"), which determine individual investment behaviors. What are the features of the strategies exhibited by individuals in the game? How does the investment contribution of participants in repeated games vary with changes in the resulting allocations?
 - 2) The author mentioned that the proportional allocation mechanism may be relatively weaker in achieving equality compared to other allocation mechanisms, potentially resulting in a situation where a minority receives a larger surplus. The proportional allocation mechanism is related to participants' historical contributions: the more one contributes, the more one receives. However, logically speaking, in a game, if participants find themselves providing less over time, they may tend to alter their current behavior of lesser contribution (similar to Win-Stay, Lose-Shift strategy). Therefore, does this imply that proportional allocation could achieve relative equality when certain strategies (contribution behaviors) are present? In such cases, under strategy use, participants' relative contributions would be closer.
 - 3) Regarding training virtual players, the aspects of the neural network involved are mentioned. However, in terms of participants' contributions, it is only briefly stated that game history is used to predict contributions in the current round. What aspects of game history are involved here? Does it refer to participants' own histories, their interaction partners', or the contribution histories of other participants? Especially when the allocation coefficient w dynamically changes, whether behaviors or contributions change and how they change remains unclear. In our view, behavior and allocation mechanisms may influence each other mutually, and further detailed descriptions of behavioral processes in dynamic allocation processes may be worth considering.
 - 4) In Exp.4, it is mentioned that games end with a 20% chance during repeated interactions, but this information is not provided in Exp.1 (line 167), which may have been overlooked by me. Additionally, when the termination probability changes, do the expected outcomes also change? Because this would affect group earnings and allocation, in other words, are the results robust to changes in these parameters?
 - 5) We noticed discrepancies in the use of symbols: r is used to represent the gain factor of the common pool in the main text, but k is used in Section S10. Similarly, p is used to represent the power coefficient at line 987, while k is used in the main text. Besides, in line 283, the mean is represented by colored dots instead of the orange dots only.
 - 6) Equ.1 is not numbered, and the description of the equation at line 135 may be ambiguous. The participants' contributions in the common pool generate enforcement gains, while the equation involves multiplying by the allocation amount c .
 - 7) Regarding the interpolating baseline, it is unclear why the allocation is determined by such an exponential form, and further clarification is needed on how the power coefficient is determined experimentally. If another form is used, does it mean that the appropriate forms can also be found experimentally to achieve similar or more ideal effects? The authors may have a better understanding of this, and perhaps this should be discussed in the corresponding section.
 - 8) Although more than 40 rounds of game data are used in the training data, the main results are primarily present in about 40 rounds. Although ideal effects under some allocation mechanisms have been observed under the current interaction rounds, it seems necessary to clarify whether these ideal effects can persist or fluctuate with an increase in game rounds. In other words, when resources are abundant and fair allocation is guaranteed temporarily, does prolonging the game rounds affect the game's outcomes? Therefore, it would be better to explicitly point this out, perhaps I have misunderstood.
 - 9) When other game parameters change, such as the number of participants in the group p or changes in the gain factor r affecting the dilemma intensity, do the main results change?
 - 10) Finally, it may be beneficial to enhance and refine the discussion section by including explanations of the limitations of this study, potential major obstacles, and promising directions for future research.
- We greatly appreciate the effort put forth in applying deep reinforcement learning to address social dilemma problems in this article. It is a pleasure reading this paper, and we hope the comments are meaningful to the authors.

(Remarks on code availability)

Reviewer #4

(Remarks to the Author)

(Remarks on code availability)

Reviewer #5

(Remarks to the Author)

Summary: The paper explores the application of deep reinforcement learning (RL) to design an allocation mechanism that encourages sustainable contributions from human participants to a common pool resource. The authors use an iterated multiplayer trust game to simulate economic interactions and train a neural network to behave like human players. This model is then used to study the influence of different mechanisms on the dynamics of receipt and reciprocation. A social planner, trained via RL, is tasked with maximizing the aggregate return to players. The study finds that the RL agent discovers a redistributive policy that promotes a large surplus and an inclusive economy, where players gain equally. The agent achieves this by conditioning its generosity on available resources and temporarily sanctioning defectors. The paper also introduces an explainable mechanism that performs similarly and is favored by players.

Recommended Citation:

1. Onymity promotes cooperation in social dilemma experiments
Z Wang, M Jusup, RW Wang, L Shi, Y Iwasa, Y Moreno, J Kurths
Science advances 3 (3), e1601444 (2017)
2. Exploiting a cognitive bias promotes cooperation in social dilemma experiments
Z Wang, M Jusup, L Shi, JH Lee, Y Iwasa, S Boccaletti
Nature communications 9 (1), 2954 (2018)
3. Communicating sentiment and outlook reverses inaction against collective risks
Z Wang, M Jusup, H Guo, L Shi, S Geček, M Anand, M Perc, CT Bauch, ...
Proceedings of the National Academy of Sciences 117 (30), 17650-17655 (2020)

Strength:

1. The findings have implications for the provisioning of public goods and the design of economic policies that aim to reduce inequality and promote sustainability.
2. The development of an explainable heuristic that approximates the RL agent's policy is a significant contribution.
3. The authors validate their model through extensive human experiments, providing a robust test of the RL mechanism's effectiveness.

Weakness:

1. While the model performs well in the context of the multiplayer trust game, it is unclear how well these findings would generalize to other types of economic interactions or real-world scenarios.
2. The training of deep RL models can be computationally expensive and time-consuming, which may limit its scalability.

Question:

1. Are there any ethical concerns regarding the use of AI to influence human behavior in economic games, especially concerning the potential for manipulation?
2. How sensitive are the results to changes in the game parameters, such as the growth factor or the maximum pool size?
3. Reproducibility concerns: the system environment itself is excellent, but not easily reproducible by researchers for further study

(Remarks on code availability)

So far, the authors do not open source the code for the submission. Also, the authors indicate that they were ultimately unable to open source all of the code because of the internal systems involved in deepmind. They will provide the human data through a python notebook with the core experiments components.

Version 1:

Reviewer comments:

Reviewer #1

(Remarks to the Author)

I have carefully reviewed the revised manuscript and the responses to the previous round of peer review. The authors have made commendable efforts to address the concerns raised in the first round, and the manuscript has been significantly improved. Below are my minor comments and suggestions for further refinement:

Given the improvements and clarifications made to the model's description, could the authors provide additional insights into how their model might be adapted or applied to other types of economic games or real-world scenarios? Specifically, how would the model perform in games with different information structures or payoff matrices?

(Remarks on code availability)

Reviewer #2

(Remarks to the Author)

Thank you for your revisions and responses to my previous comments. However, there are still some issues that need to be addressed before the manuscript can be considered for publication. See my comments below:

1. Regarding the differences between the CPR model used in this paper and the classic CPR model, the authors only provided a brief description in the last paragraph of the Discussion. However, this did not address the question why the authors used the variant of the CPR model rather than the classic one. It would be better to discuss some real-world scenarios where the model presented in this paper is more appropriate than the classic CPR model, and please highlight the advantages of this model over the classic CPR model.
2. I partly agree with the explanation of the innovations in this paper. I suggest the authors adding a brief paragraph in the discussion section to elaborate on the differences between this work and Koster et al., 2022 and summarize the main technical innovations of this paper.
3. Fig 2 is incorrect (it is currently identical to Fig 3).
4. Additionally, several issues I mentioned last time were not addressed, despite being marked as corrected in the point-by-point response. (1) In the first paragraph of the Results section, " $r=1.4$ " was not changed to " $r=0.4$ ", and several instances of 's' and 'e' were missing their subscript. (2) I agree with the explanation for using the power-law function in the "interpolating baseline." However, the statement "the power-law function is not the only possible heuristic model which could capture the mechanism discovered by the RL agent" was supposed to be included in the Discussion section but was omitted.

(Remarks on code availability)

Reviewer #4

(Remarks to the Author)

(Remarks on code availability)

Version 2:

Reviewer comments:

Reviewer #1

(Remarks to the Author)

This manuscript is based on research into trust games and public goods games, particularly focusing on trust games in repeated settings, where there is a noticeable gap in the literature concerning conditional investment behaviors. It is recommended that the authors enhance this aspect, as it would greatly benefit the understanding of the article.

(Remarks on code availability)

Reviewer #2

(Remarks to the Author)

All my comments have been addressed. The paper looks pretty nice now.

(Remarks on code availability)

Reviewer #4

(Remarks to the Author)

(Remarks on code availability)

We are grateful to all 5 reviewers for helpful comments. In what follows, we describe how we have addressed the concerns that have been raised.

Reviewer #1 (Remarks to the Author):

This MS mainly explores how to address social dilemmas by designing an allocation mechanism that can incentivize participants to make sustainable contributions to the public resource pool in situations where resources are limited. The authors first trained neural networks to simulate the behavior of human players, and then used deep reinforcement learning (RL) to train a social planner to maximize the total player return. This social planner has discovered a redistribution strategy that leads to a large surplus and inclusive economy, where players receive roughly equal benefits.

The main innovation of this study lies in the use of new tools from AI research, such as deep neural networks and deep reinforcement learning, to design and optimize resource allocation mechanisms. This method not only handles complex historical event sequences, but also maximizes social welfare within the group.

Overall, the contribution of this study lies in proposing a new perspective and method to address the social dilemma of resource allocation, which has positive implications for areas such as public interest provision, compensation, and welfare theory. The authors have successfully applied AI research tools to social science issues, which is an interdisciplinary attempt and opens up new research paths.

Although the MS is valuable, there are still a series of issues that need to be further addressed.

1) On page 3, line 135, $\Delta R_t = -\sum_i (1+r)e^{\sum_i c_i}$, this formula seems to be problematic. It seems that 'e' should be before the first '+' sign.

Thanks for spotting this typo - this has been corrected (line 134).

2) An accurate comprehension of different roles and interactions in your model is crucial for the understanding of the readers. In Figure 1A, there is an instance (Right side) where the second player receives resources from the common pool but does not reciprocate back into the resource pool, leaving a remainder of zero. Here, it is suggested that the authors explain the basic principles behind zero remainder.

Players can freely choose how much of their allocated funds to reciprocate, but if nothing is received, there is nothing to give back. We have adjusted the legend to highlight this (line 151).

3) The authors have embarked on their research utilizing a repeated multi-person trust game, an approach that diverges from previous studies. Notably, the role of the investor is not played by a specific individual participant but is instead substituted with a resource pool (Similar in form to the public goods game). Intriguingly, this resource pool exhibits the capability to make intelligent

decisions, determining the quantity of resources (depicted as blue flowers in the manuscript) to allocate to the game participants, or alternatively, to withhold them entirely. This innovative experimental design is compelling. Would the authors be able to elucidate the underlying motivation for adopting such a methodology?

In addition, the role of a resource pool is similar to that of a conditional investor in a multiplayer trust game, who can choose allocation ratios based on the game environment. Authors should explain this.

The goal of this choice is to study whether deep RL can be used to find an allocation mechanism that promotes sustainable behaviour in humans. This is what we mean when we say “Here, we asked whether a deep network could learn to dynamically allocate resources to human recipients in ways that encouraged them to sustain the common pool.” (line 78).

4) The authors recruited 640 participants. I'm wondering whether the inherent attributes of these participants (such as gender ratio, education level, etc.) might have an impact on the experimental results. I don't seem to find related research or description on this aspect in the supplementary materials. Another point is that the supplementary materials state that the final database contains 4952 participants, which needs clarification.

The tally of 4952 includes participants that were used to train the model via imitation learning (see methods) as well as those described in the main experiments reported in the paper, which are used to evaluate the properties of the trained model. We agree that it would be interesting to know more about the demographics of the participant cohort, but we did not record this information, so unfortunately we are unable to report it (this was an intentional decision as to allow players to comply with internal data protection guidance). We have now added this to the discussion as a limitation (line 441).

5) While the MS suggests implementing a more egalitarian distribution policy when resources are abundant, the discussion regarding how to adjust strategies to form the most optimal resolution in situations of extremely scarce resources is insufficient.

We weren't sure what additional discussion the reviewer would have liked to read, but we have now expanded the discussion section to describe the mechanism in more detail on line 420.

6) It would be beneficial to discuss the impact of several social factors circled around policy-making, such as social norms and demographic information, and how they might contribute to forming the optimal resource allocation strategy.

We now note in the discussion the possibility that our results may depend on demographics of the participant cohort (line 442).

7) Despite the conducted evaluation with human players, it would perhaps be enlightening to delve deeper into demographics distinguished by varying backgrounds, cultures, and age

groups (as mentioned in (4)). Further research on these specific populations might elucidate some novel and meaningful conclusions.

This is an interesting point, but as we mention above, we didn't collect this information, so we are unable to comment on this issue. However, we do now mention this as a limitation (line 442)

8) A series of experiments are designed to investigate the impact of different mechanisms on resource management through virtual environments, behavioural cloning and the training process of deep learning agents. The authors choose specific network architectures and hyperparameters such as learning rate and batch size. Can you elaborate on the reasons for choosing these parameters and their impact on the experimental results? In addition, can more specific information, such as the number of layers of the neural network, the number of neurons per layer, the choice of activation function, etc., be provided so that the reader can fully understand how the model works and what the effects are?

These details are explained in Section 7.3 and Methods Table 1. The hyperparameters were chosen by investigating train/test performances from hyperparameter sweeps. While we did not investigate the impact of any hyperparameters systematically, a wide range of values lead to good test performance.

9) In the discussion section, the authors can suggest next steps for research on how deep reinforcement learning can be further applied to promote sustainable behaviours and how the limitations in the current research can be addressed. Although, deep RL mechanisms such as the one in the paper can be used to discover a resource allocation policy and are found to be easier to understand by the human participants, will humans ultimately be able to do the same thing as these AIs?

We don't know the answer to this question, but we think that these are two slightly different issues. We investigated the interpolating mechanism in service of seeking interpretability and finding a simple description of a potentially very complicated algorithm. However, in practice, the fact that a deep RL system can be expressive and sensitive in ways that humans can't is a feature, so depending on the domain, the fact that humans cannot reproduce the 'superhuman' AI performance would be desirable.

There are a few minor suggestions for improvement:

1) On page 3, line 134, *min* should be in straight font instead of italics.

Thanks, fixed.

2) In the caption for Figure 1, it is suggested that the authors provide a more detailed explanation of what 'w' represents. Although there is a relevant explanation in line 175, it should be explained in the caption of the figure.

As far as we can see w is not mentioned in figure 1. w described the mixing parameter of a specific mechanism, whereas Figure 1A/B are general descriptions agnostic to the specifics of the mechanism, so we think explaining it there might be confusing.

3) Page 5, I recommend that the authors provide a more comprehensive explanation regarding the variable 'j' in the formula.

Thanks, fixed.

4) I recommend that the authors consider improving the resolution or clarity of Figure 2.

Thanks for spotting this. We have replaced this with a higher quality version.

5) For the sake of uniformity, it is advised that the terms 'equal', 'proportional', and 'mixed' in lines 195 and 198 are presented in italics.

We have italicized these terms on first use, but not subsequently.

6) Line 291, the phrase "This behaviour is reminiscent of Kuznets theory" might be clearer if it were "This behaviour is reminiscent of the Kuznets curve theory".

We have now included a reference to the Kuznets curve later in that sentence (line 292).

7) The references on repeated multiplayer trust games, deep reinforcement learning, and public resource management need to be moderately increased.

Thanks for these suggested changes, which we have implemented, with the exception that we didn't see the rationale for adding references without a specific reason to do so, and so have not taken action to address (7).

Reviewer #1 (Remarks on code availability):

No code provided.

We will provide code to ensure that our analysis can be reproduced when the paper is accepted.

Reviewer #2 (Remarks to the Author):

Comments of "Using deep reinforcement learning to promote sustainable human behaviour on a common pool resource problem"

This paper provides an in-depth experimental validation for promoting sustainable human behaviour in a common pool resource game. It starts by using data collected from human participants to create clones that imitate human behavior. These clones are then used to train a

reinforcement learning (RL) agent, M1. To test how generalizable the RL agent allocation mechanism is, the authors conduct Experiment 1, and find that the RL agent generated a higher surplus compared to other baselines. They investigate the properties of the RL agent further. To understand the reasons behind the effectiveness of the RL agent, the authors introduce a new baseline named the interpolating baseline and carry out Experiment 2. The findings indicate that this new baseline is as effective as the RL agent. This suggests that the success of the RL agent is due to its allocation mechanism being dependent on the size of the pool. Additionally, both human players and virtual players participate in the first two experiments. Their similar behaviors demonstrate the feasibility of using virtual players to train the RL agent. In addition, the authors made use of the data from the first two experiments to retrain new virtual players and a new RL agent, M2. Then they conduct Experiment 3 to revalidate that the new RL agent allocation mechanism excelled in maximizing surplus. Finally, to explore whether the previous outcomes are robust to the incentive structure, the authors altered the way of calculating bonuses in Experiment 4.

As a general assessment, this paper presents a very interesting conclusion. The allocation mechanism implemented by the RL agent promotes reciprocation among players more effectively than common allocation mechanisms such as fair allocation or allocation proportional to contribution. Additionally, it identifies why the RL agent allocation mechanism is effective in maximizing surplus: the mechanism depends on the size of the pool. Specifically, when the pool size is small, the allocation mechanism is close to a proportional baseline, whereas with a larger pool size, the RL agent tends to distribute more equally. This characteristic allows the agent to exclude free-riding players when resources are scarce, leaving behind players who reciprocate more. As resources increase, the allocation mechanism can then reinclude previously excluded players. Ultimately, this maximizes the total surplus while ensuring that the distribution of surpluses among players is not too unequal.

I have two main concerns about this paper.

(1) From my perspective, using a repeated multiplayer trust game framework to study the CPR problem may be inappropriate because both the objective and the equilibrium structure of the trust game differ significantly from those of the typical CPR problem.

In a traditional CPR scenario, players extract resources from a common pool without any obligation to reciprocate. The goal is to effectively manage and regulate the use of these resources to prevent depletion. However, in the described repeated multiplayer trust game, players receive endowments and decide how much they return to a common pool. This game focuses not on preventing the pool's premature depletion but on maximizing the total surplus for the players. Additionally, this game setup is more akin to a public goods game, where players receive an endowment and choose how much to contribute back. Contributions from players cannot exceed their initial endowments, and the benefits from public goods can be considered as endowments in the next round.

The reviewer is correct that our game differs slightly from the standard formulation of the CPR problem. The main difference is that players cannot choose how much to draw down from the

common pool, but instead choose how much to draw down (cash out) from their allocation. This allows us to study mechanisms that may promote sustainable behaviours in the CPR (akin to the setting of quotas). We now mention this in the discussion (line 445)

It is also true that we optimise the agent to maximise collective surplus. However, surplus is tightly linked to sustainability - because on those games where the common pool is depleted, players have no further opportunity to make a surplus.

(2) This article explores the design of an allocation mechanism using AI tools. The method involves several key steps. First, data is collected from human participants to train virtual players. These virtual players are then used to train a reinforcement learning (RL) agent. Behavioral experiments follow, comparing the RL agent with baseline models. The RL agent's mechanism effectively maximizes total surplus, leading the authors to examine its properties further. This method is based on the approach used by Koster et al., 2022, as cited in reference [21]. Compared to the study by Koster et al., 2022, the main innovation of this paper lies in devising an interpolating baseline that approximates the RL agent's allocation mechanism. This further confirms that the success of the RL agent's allocation mechanism relies on the allocation being dependent on the pool size. However, the level of innovation in this paper (i.e., devising an interpolating baseline that approximates the RL agent's allocation mechanism) may not be sufficient for Nature Communications.

To learn a mechanism we use a combination of behavioral cloning and deep RL. These are both established mechanisms in the literature and so our main contribution is not methodological. We also used these methods in Koster et al 2022, but applied to a very different problem. In that earlier paper, we studied a much simpler problem in which each decision made by the agent is independent of every other. Here, we study a much more challenging problem in which the decision made on each time step have potentially long-lasting consequences. This required the introduction of several technical innovations (such as the use of a deep RL system with activation memory) and allowed us to discover a mechanism that is much more relevant to real-world economic problems. The main contribution of our paper is methodological innovation but the results of applying established methods to a new and challenging problem.

Small questions:

(1) In the interpolating baseline, w is defined as a power-law function of pool size. Given that the RL agent was more punitive when the pool was low, similar to the proportional baseline, and made more generous offers as the common pool grew, distributing resources more equally—why wasn't a piecewise linear function considered? If w is set as a piecewise linear function dependent on pool size, with a threshold where $w=0$ below this threshold and is linearly proportional to the pool size above it, could similar results to the RL agent be achieved?

This is a nice suggestion. We suspect that a piecewise linear function would behave very similarly, as it can obviously approximate the power law function. We now mention in the discussion that the power law function is not the only possible heuristic model which could capture the mechanism discovered by the RL agent.

(2) Both Experiment 2 and Experiment 3 indicated that the interpolating baseline generated a higher surplus than the proportional baseline. Given these findings, I'm curious why Experiment 4 chose to compare RL agent M2 with the proportional baseline instead of the interpolating baseline. Like in Experiment 3, it should be compared with the interpolating baseline to test whether the results they obtained may be due to the incentive structure.

We were also considering whether to do Experiment 4 with the interpolating baseline but opted not to (it is a significant investment to do so) because Experiment 3 and 4 are two different 'branches' exploring the key results in different ways. The point of experiment 4 is to ask whether the agent's policy goes out of distribution when participants play the game for longer; it shows that this is not the case, which establishes the generality of our solution. It also explores whether the results are robust to the variation in the game's termination rule (and the participants' belief about the game length), as the specifics of this rule are of great interest to behavioural economics.

We did not conduct experiment 4 to make a claim about relative strength to the interpolating baseline (ie. we are not claiming our agent is best), because we have already shown that the agent and interpolating baseline are equivalent; we do not think it is necessary to show this again (and we make no strong claims related to this).

(3) In the first paragraph of the 'Results' section, there is a subscript missing from "e" in the expression " $\sum_i e \leq R_t$ ". The expression for ΔR_t is incorrect. In " $\sum_i \sum_t s$ ", "s" is also missing a subscript. Additionally, both this section and the caption for Figure 1 mention " $r=1.4$ ", but it should correctly be " $r=0.4$ ".

Thanks, we have corrected these typos.

(4) In the third paragraph on page 7, there is an error where Fig. 2B is mistakenly referred to as Fig. 2A. Similarly, on page 9, Fig. 4A is incorrectly cited as Fig. 3A.

Thank you for spotting these typos, which have now been corrected.

(5) Could you provide a more detailed description of the superimposed colored histograms in Fig. 2D? My confusion arises from the observation that there are fewer bars than 40, which does not seem to correspond directly to the values on the horizontal axis. How are these histograms constructed, and what does each bar represent in relation to the axis values?

Thank you for this note, indeed for all plots that overlay a histogram over a scatterplot, the histogram bins span 2 values on the x axis, this was purely a decision for visual balance given the size of the plots. We have now added extra explanation to ensure that readers can understand what is shown (line 235)

(6) The term “game” is frequently used in the text to refer to all the rounds played by a group, but the definition of “game” is not clearly stated, which leads to confusion about the scope of what “game” refers to. It would be better to include a clear definition of 'game' in the manuscript.

This is defined on line 130.

(7) On page 17, screenshots of the experiment show that each participant is assigned a number. Is this numbering fixed? For example, does the player labeled as Number 3 remain as Number 3 throughout the entire game on a specific participant's interface?

Each player always sees themselves as player 1. In experiments 1-3, players only play a single game. In experiment 4 the player slots remain fixed. This is intentional as to give continuity to players over the 3 games.

(8) On page 19, the 'Mechanisms' section mentions that the data used to train virtual players is collected under two baselines. However, this description conflicts with the information on page 22 in section 7.2 “Initial data collection”, where it is stated that the training set also includes play under a prototype neural mechanism M0. I would suggest that the authors to make the descriptions consistent.

Thanks - this has been clarified (line 675).

(9) In the first line on page 20, the text in parentheses following “2 Baselines” should not include “M1”.

Thank you, corrected.

Reviewer #3 (Remarks to the Author):

The authors have innovatively applied artificial intelligence as a tool to address the social dilemma of resource allocation. Specifically, they have utilized deep reinforcement learning (RL) to design a resource allocation mechanism in a multiplayer trust game, discussing how this mechanism can incentivize participants to sustain the sustainability of a shared pool. This represents a natural extension of applying AI research tools to studies on resource allocation. By comparing this mechanism with other allocation mechanisms, the authors have found its effectiveness in achieving collective benefits while also considering distribution equality. Through the demonstration of heuristic mechanisms, they have revealed typical allocation characteristics of RL mechanisms, showing a tendency towards more equal distribution when resources are more abundant. Although most results are obtained through comprehensive simulations, the motivation is clear, and the study holds significant value for research into cooperative behavior across various disciplines. Yet, we believe some improvement can be made.

1) Throughout the entire paper, the main focus and results revolve mainly around the allocation mechanism, but the description of participants' strategic behavior could possibly be more detailed. Particularly, while the authors consider the outcomes of virtual participants in the game, the strategic characteristics exhibited by these participants in the game are still unclear. In repeated games, there can be various types of strategies involved (See "Repeated prisoner's dilemma games in multi-player structured populations with crosstalk"; "Evolution of cooperation with nonlinear environment feedback in repeated public goods game"), which determine individual investment behaviors. What are the features of the strategies exhibited by individuals in the game? How does the investment contribution of participants in repeated games vary with changes in the resulting allocations?

The reviewer is right that our analysis focuses on the mechanism, although the human behaviour and that of the mechanism are of course intertwined. The main human behavioural summary data we show is in Figure 2E and 3D, where we show how reciprocation depends on endowment allocation over time. We agree that this is rather impoverished; we have now expanded this by adding a new figure offering a fuller description of how human reciprocation depends on allocation (Fig. S11). This figure also has the virtue of highlighting the strong overlap in human and behavioural clone behaviour.

We agree that it would be possible to dig more carefully into how players' behaviour is mutually interdependent over time, but drawing strong conclusions is very hard, because unlike in the PD experiments the reviewer is referring to, players' opportunity to contribute depends on their allocation, and so is not independent of their behaviour (or that of others) on past trials. This means that any depiction of (say) how cooperative behaviour of player i depends on the past reciprocation of that player or other players $_i$ is heavily confounded by the behaviour of the mechanism itself. This is why we chose instead to focus on analysis of the mechanism.

2) The author mentioned that the proportional allocation mechanism may be relatively weaker in achieving equality compared to other allocation mechanisms, potentially resulting in a situation where a minority receives a larger surplus. The proportional allocation mechanism is related to participants' historical contributions: the more one contributes, the more one receives. However, logically speaking, in a game, if participants find themselves providing less over time, they may tend to alter their current behavior of lesser contribution (similar to Win-Stay, Lose-Shift strategy). Therefore, does this imply that proportional allocation could achieve relative equality when certain strategies (contribution behaviors) are present? In such cases, under strategy use, participants' relative contributions would be closer.

In our game, participants cannot reciprocate more than they receive. Thus, participants who receive very little are obliged to give very little - they do not have, as the reviewer implies, any opportunity to alter their behaviour. That is why proportional leads to a relatively unequal distribution of resources over players.

3) Regarding training virtual players, the aspects of the neural network involved are mentioned. However, in terms of participants' contributions, it is only briefly stated that game history is used

to predict contributions in the current round. What aspects of game history are involved here? Does it refer to participants' own histories, their interaction partners', or the contribution histories of other participants? Especially when the allocation coefficient w dynamically changes, whether behaviors or contributions change and how they change remains unclear. In our view, behavior and allocation mechanisms may influence each other mutually, and further detailed descriptions of behavioral processes in dynamic allocation processes may be worth considering.

A recurrent neural network maps its hidden state at time $t-1$ onto its current hidden state via a set of recurrent weights. This serves as a memory mechanism, allowing it to use the contribution history to decide how to allocate resources. However, we do not explicitly tell the network how to use this contribution history - it learns to do it autonomously during the optimization process. It is absolutely right that contributions and allocations influence one another; this is part of what makes the problem hard for the neural network to learn.

4) In Exp.4, it is mentioned that games end with a 20% chance during repeated interactions, but this information is not provided in Exp.1 (line 167), which may have been overlooked by me. Additionally, when the termination probability changes, do the expected outcomes also change? Because this would affect group earnings and allocation, in other words, are the results robust to changes in these parameters?

The fixed termination probability was introduced to ensure that players cannot strategically hoard resources during the game and then cash out immediately before (known) termination point. This is equivalent to players having a fixed discount function. We actually used two approaches to this problem: (1) all games lasted 40 rounds, but players were informed that their contributions would only count up to a given trial, determined by an equiprobable termination point (experiments 1-3) and (2) all games genuinely terminated with a probability $p = 0.2$ (after a 25 fixed trials) on each trial (see line 394). These did not yield different results. Thus, whilst we did not vary the termination probability, we think it unlikely that it had a major impact on the results.

5) We noticed discrepancies in the use of symbols: r is used to represent the gain factor of the common pool in the main text, but k is used in Section S10. Similarly, p is used to represent the power coefficient at line 987, while k is used in the main text. Besides, in line 283, the mean is represented by colored dots instead of the orange dots only.

Thanks for spotting this, which has been corrected in both figure and legend.

6) Equ.1 is not numbered, and the description of the equation at line 135 may be ambiguous. The participants' contributions in the common pool generate enforcement gains, while the equation involves multiplying by the allocation amount c .

Thanks - fixed.

7) Regarding the interpolating baseline, it is unclear why the allocation is determined by such an exponential form, and further clarification is needed on how the power coefficient is determined experimentally. If another form is used, does it mean that the appropriate forms can also be found experimentally to achieve similar or more ideal effects? The authors may have a better understanding of this, and perhaps this should be discussed in the corresponding section.

We fit the exponent of the power function to the data to obtain the best-fitting coefficient. The point of this exercise was not to find the best functional form for the approximation, but to show that an approximation is possible. There are no doubt other simple functions (such as piecewise linear, as suggested by another reviewer) that would fit the data, but whether or not this function fit better would not change our conclusions, so we prefer not to pursue this further.

8) Although more than 40 rounds of game data are used in the training data, the main results are primarily present in about 40 rounds. Although ideal effects under some allocation mechanisms have been observed under the current interaction rounds, it seems necessary to clarify whether these ideal effects can persist or fluctuate with an increase in game rounds. In other words, when resources are abundant and fair allocation is guaranteed temporarily, does prolonging the game rounds affect the game's outcomes? Therefore, it would be better to explicitly point this out, perhaps I have misunderstood.

In experiment 4 we have games of different lengths, but obtained highly conserved results; thus, we think it unlikely that game length makes a difference.

9) When other game parameters change, such as the number of participants in the group p or changes in the gain factor r affecting the dilemma intensity, do the main results change?

We did not manipulate these variables, so we do not know this, but we agree with the reviewer that this would be an excellent path to pursue in future research.

10) Finally, it may be beneficial to enhance and refine the discussion section by including explanations of the limitations of this study, potential major obstacles, and promising directions for future research.

We have made a number of additions to the discussion in response to points raised during the rebuttal; these are included in a new "limitations" paragraph.

We greatly appreciate the effort put forth in applying deep reinforcement learning to address social dilemma problems in this article. It is a pleasure reading this paper, and we hope the comments are meaningful to the authors.

Thank you very much for your helpful comments.

Reviewer #4 (Remarks to the Author):

Thank you very much for your input to the review process.

Reviewer #5 (Remarks to the Author):

Summary: The paper explores the application of deep reinforcement learning (RL) to design an allocation mechanism that encourages sustainable contributions from human participants to a common pool resource. The authors use an iterated multiplayer trust game to simulate economic interactions and train a neural network to behave like human players. This model is then used to study the influence of different mechanisms on the dynamics of receipt and reciprocation. A social planner, trained via RL, is tasked with maximizing the aggregate return to players. The study finds that the RL agent discovers a redistributive policy that promotes a large surplus and an inclusive economy, where players gain equally. The agent achieves this by conditioning its generosity on available resources and temporarily sanctioning defectors. The paper also introduces an explainable mechanism that performs similarly and is favored by players.

Recommended Citation:

1. Onymity promotes cooperation in social dilemma experiments
Z Wang, M Jusup, RW Wang, L Shi, Y Iwasa, Y Moreno, J Kurths
Science advances 3 (3), e1601444 (2017)
2. Exploiting a cognitive bias promotes cooperation in social dilemma experiments
Z Wang, M Jusup, L Shi, JH Lee, Y Iwasa, S Boccaletti
Nature communications 9 (1), 2954 (2018)
3. Communicating sentiment and outlook reverses inaction against collective risks
Z Wang, M Jusup, H Guo, L Shi, S Geček, M Anand, M Perc, CT Bauch, ...
Proceedings of the National Academy of Sciences 117 (30), 17650-17655 (2020)

We have included citations to these papers when discussing auxiliary mechanisms, including communication, that may promote sustainable cooperation (line 61).

Strength:

1. The findings have implications for the provisioning of public goods and the design of economic policies that aim to reduce inequality and promote sustainability.
2. The development of an explainable heuristic that approximates the RL agent's policy is a significant contribution.
3. The authors validate their model through extensive human experiments, providing a robust test of the RL mechanism's effectiveness.

Thanks for these comments.

Weakness:

1. While the model performs well in the context of the multiplayer trust game, it is unclear how well these findings would generalize to other types of economic interactions or real-world scenarios.

We agree, and have now emphasised this more firmly in the discussion (line 449).

2. The training of deep RL models can be computationally expensive and time-consuming, which may limit its scalability.

We agree with the general notion of this comment but we think this is a point slightly too broad to discuss in the paper. General-purpose machine learning methods tend to scale very well in one sense (as seen for example with the success of large generative models, e.g. LLMs) because their execution is relatively cheap. Deep RL methods like ours have already been applied successfully to domains where training is much more compute intensive than in our domain (e.g. 3D games). Of course 'scaling' this method to a real world problem could be very costly depending on the size of datasets, richness of features and size of networks and training methods (if just dealing with numbers representing economic quantities or transactions it may also be very cheap compared to multi-modal data).

However, our specific implementation is fully differentiable and very cheap to optimize, so the critique of being costly (which is usually correct in AI), specifically, does not apply to our study and would make the discussion seem disconnected from the content of the study.

Question:

1. Are there any ethical concerns regarding the use of AI to influence human behavior in economic games, especially concerning the potential for manipulation?

We agree that there is always a risk of manipulation in human-AI interactions. We now mention this in the discussion.

2. How sensitive are the results to changes in the game parameters, such as the growth factor or the maximum pool size?

We did not manipulate these variables, so we do not know this, but we agree with the reviewer that this would be an excellent path to pursue in future research.

3. Reproducibility concerns: the system environment itself is excellent, but not easily reproducible by researchers for further study

Whilst we are not able to provide the code that was used to train the model, we hope that our descriptions in the methods provide readers with all the information they need to reproduce our results.

Reviewer #5 (Remarks on code availability):

So far, the authors do not opensource the code for the submission. Also, the authors indicate that they were ultimately unable to open source all of the code because of the internal systems involved in deepmind. They will provide the human data through a python notebook with the core experiments components.

Correct. This is exactly the procedure we followed for <https://www.nature.com/articles/s41562-022-01383-x>

We want to thank the reviewers for the opportunity to improve our manuscript in one more iteration, please see our responses below.

REVIEWER COMMENTS

Reviewer #1 (Remarks to the Author):

I have carefully reviewed the revised manuscript and the responses to the previous round of peer review. The authors have made commendable efforts to address the concerns raised in the first round, and the manuscript has been significantly improved. Below are my minor comments and suggestions for further refinement:

Given the improvements and clarifications made to the model's description, could the authors provide additional insights into how their model might be adapted or applied to other types of economic games or real-world scenarios? Specifically, how would the model perform in games with different information structures or payoff matrices?

We are pleased the reviewer appreciated our revisions. We agree that adapting this methods across a wider set of games/scenarios is interesting, we added the following to the discussion (in the penultimate paragraph):

'The machine learning architecture and pipeline used to tackle this specific problem is very general. The model of human behavior makes no assumptions about the structure of the game, its inputs and outputs or what humans should aim to maximize. The mechanism, being equipped with the ability to retain memories within an episode, is similarly flexible. While we varied different factors of the game (e.g. Fig S2 explores longer games), the pipeline itself should be amenable to yet more radical changes in the game structure (e.g. games more focused on risk or coordination) or input space (e.g. equipping the agents with convolutional neural networks should enable the processing of video games from pixels). An exciting challenge would be to find real world applications in which both initial data is available and in which exploring interventions via a mechanism is safe (e.g. designing auctions, recommendation algorithms, managing queues in an amusement park, setting incentives for contributions in virtual communities). The inclusion of memory to accommodate rich dynamics within long episodes is a key technical advance over Koster et al 2022 (23). Retaining activations over the whole episode makes the setup more compatible with a much more general class of games, rather than a narrow set of stylized strategic games.'

Reviewer #2 (Remarks to the Author):

Thank you for your revisions and responses to my previous comments. However, there are still some issues that need to be addressed before the manuscript can be considered for publication. See my comments below:

1. Regarding the differences between the CPR model used in this paper and the classic CPR model, the authors only provided a brief description in the last paragraph of the Discussion. However, this did not address the question why the authors used the variant of the CPR model rather than the classic one. It would be better to discuss some real-world scenarios where the model presented in this paper is more appropriate than the classic CPR model, and please highlight the advantages of this model over the classic CPR model.

We hope the paragraph added above partially addresses this issue. Our claim is not that our particular version of the CPR is of unique appropriateness or realism, but rather that our overall method can be applied to any variation of strategic games. The reason we implemented the CPR starting with an offer of the mechanism is that it was conceptually starting with the 'Trust Game', and the fact that in an iterated trust game the players form relationships that rely on memory of past actions. Given that long episodes are always players taking turns between offers and reciprocations, the order in which these happen (mechanism first or player first) only matters on the first trial. We adjusted the final paragraph to express this now, clearly addressing the 'why'.

'Secondly, we note that the game employed here is somewhat different from the classic CPR problem, in that rather than drawing down freely from the common pool, players decide what to keep or reciprocate from a pre-set allocation, decided by the mechanism designer. This follows the logic of an iterated trust game. This formulation highlights the temporal dynamics within an episode in which players need to build a relationship over time. This design aligns with the design of the mechanism that can retain activations within an episode.'

2. I partly agree with the explanation of the innovations in this paper. I suggest the authors adding a brief paragraph in the discussion section to elaborate on the differences between this work and Koster et al., 2022 and summarize the main technical innovations of this paper.

We included a discussion of this in the above paragraph (R1 response) where it fits well into the flow, but we want to avoid too much repetition with the text addressing the previous paper in the introduction.

3. Fig 2 is incorrect (it is currently identical to Fig 3).

We apologize for this manual error, this has been fixed now.

4. Additionally, several issues I mentioned last time were not addressed, despite being marked as corrected in the point-by-point response. (1) In the first paragraph of the Results section, "r=1.4" was not changed to "r=0.4", and several instances of 's' and 'e' were missing their subscript. (2) I agree with the explanation for using the power-law function in the "interpolating baseline." However, the statement "the power-law function is not the only possible heuristic

model which could capture the mechanism discovered by the RL agent" was supposed to be included in the Discussion section but was omitted.

We thank the reviewer for pointing this out. Similar as above this was an error of uploading the wrong version of the paper, this has been corrected now (we fixed the subscripts in the first results paragraph and updated $r = 0.4$ in two places).

Reviewer #1 (Remarks to the Author):

This manuscript is based on research into trust games and public goods games, particularly focusing on trust games in repeated settings, where there is a noticeable gap in the literature concerning conditional investment behaviors. It is recommended that the authors enhance this aspect, as it would greatly benefit the understanding of the article.

Thanks for this comment. However, we are not entirely clear what is meant by the “literature concerning conditional investment behaviours”. Perhaps the reviewer is referring to an evolutionary game theory literature that focusses on the emergence of trust in networks of agents playing investment games? The only reference we could find to the term the reviewer uses is by a single paper by Liu et al 2022, which comes from this tradition. Here, “conditional investment” refers to conditioning a decision to invest on a costly observation about who is trustworthy and who is not. We recognise that this literature, whilst not directly relevant to the mechanism design problem we discuss, considers trust as a multi-agent problem and so we have included a citation to the Liu paper in the introduction.

Reviewer #2 (Remarks to the Author):

All my comments have been addressed. The paper looks pretty nice now.

Thanks for your review.

Reviewer #4 (Remarks to the Author):

Additional review:

Upon reviewing the authors' responses, I have a couple of points to highlight:

While the authors address some aspects of innovation by emphasizing the complexity of the problem and their introduction of technical innovations such as a deep reinforcement learning system with activation memory, I believe their claims may still fall short in fully satisfying the reviewer's concerns (reviewer 2) about the level of innovation.

Thanks for cross-referencing with other reviewers' comments, but R2 seems to disagree and now states that his / her comments have been addressed, so we don't exactly know which changes to make to address this comment.

The acknowledgment that their main contribution is not methodological could potentially detract from the paper's impact, particularly given the expectations of Nature Communications.

We are not sure how to address this comment.

Additionally, a more thorough comparison with existing literature to clarify how their contributions distinctly advance the field would strengthen their position.

Without knowing exactly which literature the reviewer is referring to here, we don't know how to address this comment.

On a more positive note, the authors have provided a thorough response to Reviewer 5's feedback. Their explanations regarding the technical innovations and how they relate to real-world economic problems have been articulated clearly.

Thank you for considering my feedback.

Thank you for your review